# Gradient Variance Reveals Failure Modes in Flow-Based Generative Models

**Teodora Reu**
University of Oxford
teodora.reu@cs.ox.ac.uk

**Sixtine Dromigny**
University of Oxford

**Michael Bronstein**
University of Oxford

**Francisco Vargas**
Xaira Therapeutics

## Abstract

Rectified Flows learn ODE vector fields whose trajectories are straight between source and target distributions, enabling near one-step inference. We show that this straight-path objective reveals fundamental failure modes: under deterministic training, low gradient variance drives memorization of arbitrary training pairings, even when interpolant lines between training pairs intersect. To analyze this mechanism, we study Gaussian-to-Gaussian transport and use the loss gradient variance across stochastic and deterministic regimes to characterize which vector fields optimization favors in each setting. We then show that, in a setting where all interpolating lines intersect, applying Rectified Flow yields the same specific pairings at inference as during training. More generally, we prove that a memorizing vector field exists even when training interpolants intersect, and that optimizing the straight-path objective converges to this ill-defined field. At inference, deterministic integration reproduces the exact training pairings. We validate our findings empirically on the CelebA dataset, confirming that deterministic interpolants induce memorization, while the injection of small noise restores generalization.

## 1 Introduction

The current state of the art in generative modeling involves learning dynamics between a known source distribution—such as a standardized Gaussian—and a target distribution from which many samples are available. When the learned dynamics are based on an ordinary differential equation (ODE), the model effectively learns a vector field connecting the two distributions [Lipman et al., 2022, Tong et al., 2023b,a]. When the learned dynamics are based on a stochastic differential equation (SDE), either the score function [Ho et al., 2020, Song et al., 2020], or both the score and the vector field [Albergo et al., 2023], are learned, depending on the choice of SDE. Ultimately, an SDE can be rewritten as an ODE—referred to in the literature as the *probability flow ODE* [Song et al., 2020].

These methods perform very well in practice across a wide range of data modalities: images [Ho et al., 2022a, Balaji et al., 2022, Rombach et al., 2022], video [Ho et al., 2022b, Wang et al., 2024b, Zhou et al., 2022], audio [Huang et al., 2023, Kong et al., 2020, Liu et al., 2023, Ruan et al., 2023], and molecular data [Hoogeboom et al., 2022, Xu et al., 2022]. However, their main limitations are the high computational cost of the generation of samples (as integration over an ODE is required) [Song et al., 2020, Tong et al., 2023b, Albergo et al., 2023], and their inability to learn optimal transport maps, which are often essential for unpaired data translation tasks [Shi et al., 2024].

To address computational inefficiency, new models have been proposed. Consistency models accelerate sampling for score-based approaches [Song et al., 2023, Salimans and Ho, 2022, Kim et al., 2023], while Rectified Flows [Liu, 2022, Bansal et al., 2024, Lee et al., 2024] aim to learn straight

| Does low $\text{Var}(\nabla_\theta L_{\text{MC}})$ translate into learning straight or optimal vector fields? | Why not? | Consequences on ReFlow |
|---|---|---|
| $\rightarrow$ No, in deterministic regimes (e.g. ReFlow(k>1)) $\rightarrow$ Yes, in stochastic regimes (e.g. CFM, SBM) | Under a deterministic regime, points where interpolating lines intersect are measure-zero, making $(x_t, t) \rightarrow (x_0, x_1)$ injective over the training samples. | ReFlow tends to memorize arbitrary deterministic pairings in the training data, without improving the transport couplings (Proposition 2). |

Figure 1: Intuition behind the main results.

vector fields that can be integrated in a single step. Rectified Flows iteratively update couplings and retrain vector fields to straighten transport paths, but repeated rectifications can accumulate errors, and it remains theoretically unclear whether one or two rectifications suffice under general conditions. Input-Convex Neural Networks (ICNN)-based parameterizations can, in theory, guarantee optimal couplings for noiseless interpolants, but are difficult to optimize in practice [Makkuva et al., 2020, Huang et al., 2020].

Schrödinger Bridge Matching (SBM) [Shi et al., 2024, Peluchetti, 2023, De Bortoli et al., 2024] extends these ideas by learning both forward and backward vector fields using noisy interpolants, approximating entropic optimal transport. This bidirectional approach can improve stability and mitigate error accumulation, but requires training two networks per iteration.

In this paper, we address fundamental limitations of iterative generative models by analyzing how gradient variance reveals suboptimality in learned vector fields. Our theoretical and empirical study uncovers why standard neural architectures struggle to represent even simple transports and how repeated rectifications can lead to memorization rather than improvement.

**Contributions**  In Section 3, we investigate the question "When is gradient variance informative?" in the Gaussian-to-Gaussian setting. We show that the answer depends on the training regime (stochastic or deterministic), and that in deterministic regimes, ill-defined vector fields that memorize the training pairs may emerge. Figure 6 provides a central illustration of this phenomenon. Additionally, we derive bounds for the loss and gradient variance under both regimes and demonstrate that our empirical observations align with the theoretical predictions from Proposition 1.

In Section 4, we extend these results from the Gaussian-to-Gaussian case to general finite training datasets within the ReFlow paradigm. We prove (Proposition 2) that there exists a minimizer on the finite dataset that reproduces the training pairings. Moreover, due to the deterministic nature of the integration method, the model can reproduce these exact pairings at inference time by effectively "jumping over" intersections (Remark 1). We validate our theoretical findings empirically on both a Mixture of Gaussians and the CelebA dataset in Section 4.2, and Section 4.3.

## 2   Background

**Notation.**  Let $\mathbb{R}^d$ denote $d$-dimensional Euclidean space. We use $\mathcal{P}(\mathbb{R}^d)$ for the set of probability distributions on $\mathbb{R}^d$. The source and target distributions are $\pi_0$ and $\pi_1$, with $X_0 \sim \pi_0$ and $X_1 \sim \pi_1$. A transport map is $T : \mathbb{R}^d \rightarrow \mathbb{R}^d$, and $T_\# \pi_0$ denotes the pushforward of $\pi_0$ by $T$. Interpolants are written $X_t = I(X_0, X_1, t)$, with $t \in [0, 1]$. We write $\mathbb{E}[\cdot]$ for expectation, $\text{Var}[\cdot]$ for variance, and $\nabla_\theta$ for gradients with respect to parameters $\theta$. Bold symbols (e.g., $\mathbf{x}$) denote vectors. All other notation is defined in context. For $k \in \mathbb{R}$, $R_{k\circ}$ denotes a $d \times d$ rotation matrix corresponding to a counterclockwise rotation by $k$ degrees in the plane of interest. All other notation is defined in context.

**Optimal Transport and Entropy-Regularized OT.**  Let $\pi_0, \pi_1$ be probability measures on $\mathbb{R}^d$. The Monge problem [Monge, 1781] seeks a transport map $T$ minimizing:

$$\inf_T \int \|T(x) - x\|^2 \, d\pi_0(x) \quad \text{s.t.} \quad T_\# \pi_0 = \pi_1, \tag{1}$$

where $T_{\#}\pi_0$ denotes the pushforward of $\pi_0$ under $T$. The Kantorovich [Kantorovitch, 1958] relaxation introduces couplings $\pi \in \Pi(\pi_0, \pi_1)$ and solves:

$$\mathcal{W}_2^2(\pi_0, \pi_1) = \inf_{\pi \in \Pi(\pi_0, \pi_1)} \mathbb{E}_{(X_0, X_1) \sim \pi}[\|X_1 - X_0\|^2], \tag{2}$$

where $\Pi(\pi_0, \pi_1)$ denotes the set of joint distributions with marginals $\pi_0$ and $\pi_1$. Entropy-regularized Optimal Transport (eOT) adds a Kullback–Leibler divergence penalty $\mathcal{W}_2^\epsilon(\pi_0, \pi_1) = \inf_{\pi \in \Pi(\pi_0, \pi_1)} \mathbb{E}_\pi[\|X_1 - X_0\|^2] + \epsilon D_{\mathrm{KL}}(\pi \| \pi_0 \otimes \pi_1)$.

When $\pi_0$ is absolutely continuous, the Monge and Kantorovich problems admit the same deterministic optimal plan. A dynamic formulation describes OT as evolving a path $\{X_t\}_{t \in [0,1]}$ connecting $X_0 \sim \rho_0$ and $X_1 \sim \rho_1$. For convex costs, the optimal path is given by the straight-line interpolant $X_t = (1-t)X_0 + tX_1$ [McCann, 1997].

**Conditional Flow Matching.** Given the interpolants $X_t = I(X_0, X_1, t) = \alpha_t X_0 + \beta_t X_1 + \gamma_t \epsilon$ where $\epsilon \sim \mathcal{N}(0, I)$, the CFM objective Lipman et al. [2022] is:

$$\mathcal{L}_{CFM}(v) = \mathbb{E}_{t \sim \mathcal{U}(0,1), (X_0, X_1) \sim \pi, \epsilon}[\|(X_1 - X_0) - v(X_t, t)\|^2] \tag{3}$$

The learned vector field $v$ generates flows via the ODE:

$$\frac{d}{dt}X_t = v(X_t, t) \tag{4}$$

**Rectified Flows (or ReFlow).** Liu et al. [2022] propose an iterative procedure to straighten transport paths. At each iteration $k$, a vector field $v^{(k)}$ is trained using the CFM loss $\mathcal{L}_{CFM}$ on the current coupling $(X_0^{(k)}, X_1^{(k)})$. The updated coupling is then generated via $X_1^{(k+1)} = \text{ODE-Solve}[v^{(k)}](X_0^{(k)}, t = 1)$, and the process repeats until convergence. A coupling $(X_0^{(k)}, X_1^{(k)})$ is considered straight if

$$\mathbb{E}\left[X_0^{(k)} - X_1^{(k)} \mid X_t^{(k)} = x\right] = X_0^{(k)} - X_1^{(k)}. \tag{5}$$

where $X_t^{(k)} = (1-t)X_0^{(k)} + tX_1^{(k)}$. A coupling is considered straight if, for deterministic couplings $(X_0, X_1)$, the mapping $(X_t, t) \mapsto (X_0, X_1)$ is injective. Although some works claim one or two rectifications suffice to obtain straight couplings [Lee et al., 2024] , and [Bansal et al., 2024] (in the case of isotropic Gaussian), there are no theoretical results guaranteeing it for any continuos source and target distributions.

**Gradient Variance.** We begin by rewriting the CFM objective from Equation 3:

$$L(v, T, I) = \mathbb{E}_{X_0 \sim \pi_0}\left[\|T(X_0) - X_0 - v(X_t, t)\|^2\right], \quad X_t = I(X_0, T(X_0), t), \tag{6}$$

where $T$ is a (possibly random or deterministic) map satisfying $T_{\#}\pi_0 = \pi_1$. A Monte Carlo approximation of this loss, using samples $\{X_0^{(s)}\}_{s=1}^S$, is given by:

$$L_{\mathrm{MC}}(v, T, I) = \frac{1}{S}\sum_{s=1}^{S}\left\|T\left(X_0^{(s)}\right) - X_0^{(s)} - v\left(X_t^{(s)}, t\right)\right\|^2, \quad X_t^{(s)} = I(X_0^{(s)}, T(X_0^{(s)}), t)). \tag{7}$$

The gradient variance will have the following formulation:

$$Var[\nabla_\theta L_{\mathrm{MC}}(v, T, I)] = Var\left[\nabla_\theta \frac{1}{S}\sum_{s=1}^{S}\left\|T\left(X_0^{(s)}\right) - X_0^{(s)} - v\left(X_t^{(s)}, t\right)\right\|^2\right]. \tag{8}$$

**Memorization and Generalization.** Memorization and Generalization have been extensively explored in the field of generative modeling [Bamberger et al., 2025, Buchanan et al., 2025, Somepalli et al., 2023, Ren et al., 2024, Rahman et al., 2025, Wang et al., 2024a, Stein et al., 2023] with privacy concerns [Ghalebikesabi et al., 2020] and copyright infringement [Cui et al., 2023] being key issues that would result from such frameworks. Our setting differs from most prior memorization studies by assuming that each training target has a deterministic counterpart in the source, as in Rectified

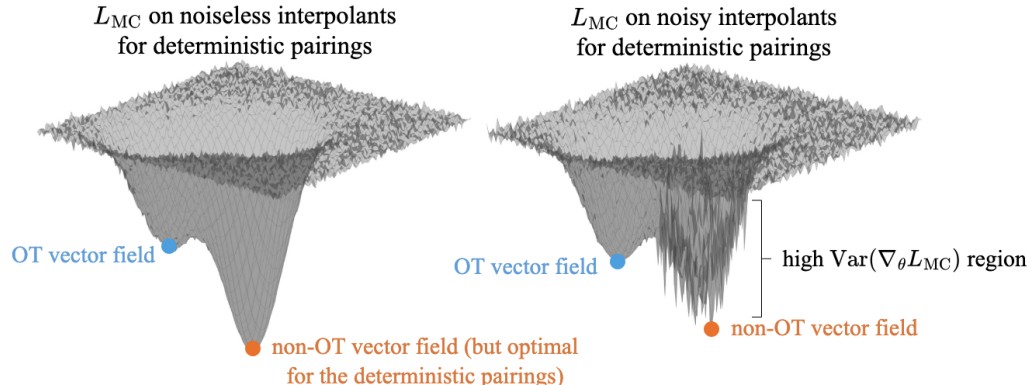

$L_{\mathrm{MC}}$ on noiseless interpolants
for deterministic pairings

$L_{\mathrm{MC}}$ on noisy interpolants
for deterministic pairings

OT vector field

OT vector field

high $\mathrm{Var}(\nabla_\theta L_{\mathrm{MC}})$ region

non-OT vector field (but optimal
for the deterministic pairings)

non-OT vector field

Figure 2: Schematic representing a hypothetical loss ($L_{\mathrm{MC}}$) landscape. The gradient variance of the loss acts as an indicator of solution quality. The schematic illustrates how the variance of the loss gradient reveals information about the optimality of the vector field under different interpolant types.

Flows, rather than sampling source–target pairs independently as is standard in the CFM literature. As a consequence, the appropriate notion of memorization changes: instead of asking whether a model starting from arbitrary source points reproduces pairs from the training dataset, we consider a deterministic training set of pairs $(X_0, X_1)$ and say that memorization occurs if the ODE integration of the learned vector field from a training source $X_0$ returns its paired target $\hat{X}_1 = X_1$ at inference, effectively reproducing the training coupling. This reframing highlights the central question in rectified-flow training: what value is added by learning a new vector field if deterministic integration simply recovers the original pairings, i.e., amounts to memorization rather than improved transport structure or generalization.

## 3 Gradient Variance of Gaussian-to-Gaussian transport

Analyzing $Var[\nabla_\theta L_{\mathrm{MC}}(v, T, I)]$ across different *choices of pairings* $T$ (e.g. random vs. optimal vs. straight), *interpolants* $I$ (noiseless vs. stochastic), and *vector-field classes* $v_\theta$ provides insight into which solutions are favored by the loss landscape. For clarity, we distinguish vector fields that are optimal in the OT sense from those that are optimal for a specific pairing (minimizing error given that pairing); we refer to the latter as pair-optimal. Although a loss may have multiple minima, optimization often prefers those with lower gradient variance, even if they are suboptimal in terms of transport cost. Figure 2 illustrates this effect: for deterministic couplings and noiseless interpolants, the lowest variance and loss minimizer can be non-OT, while introducing stochasticity can shift preference toward more optimal solutions.

Because Gaussian-to-Gaussian OT is analytically tractable (Lemma 1), the exact gradient variance can be derived for a broad set of vector-field classes, interpolants, and pairing schemes in Proposition 1. To broaden the comparison, a rotational OT (rOT) field is included that first rotates the source and then maps it to the target; these trajectories are straight for all angles except the degenerate $180°$ case,

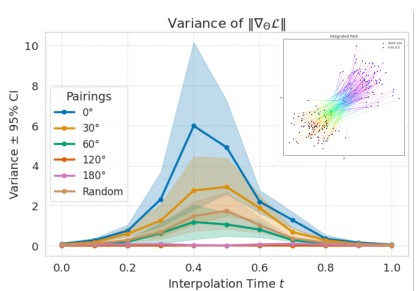

Figure 3: Gradient variance over time for a vector field rotating by $120°$, under various pairing types including rotated and random couplings. In the top right, the two Gaussians are shown along with sample trajectories, color-coded by integration time. Optimal pairings exhibit the highest variance, while random pairings do not yield the maximum variance.

which is not well-defined. This controlled setting enables closed-form analysis of variance across configurations and direct alignment with empirical measurements.

In Lemma 1, we derive this closed-form expression for the optimal vector field between two Gaussian distributions with different means and covariances, and we show that a multilayer perceptron (MLP) cannot exactly represent it without error, motivating the use of the closed-form parameterization rather than an MLP surrogate.

**Lemma 1.** *Let $X_0 \sim \mathcal{N}(0, \boldsymbol{I}_d)$ and $X_1 \sim \mathcal{N}(\mu, \boldsymbol{M}_d)$, where $\boldsymbol{M}_d$ is a positive definite and symmetric matrix. The OT vector field is given by*

$$\hat{v}_{OT}(X_t, t, \hat{\boldsymbol{\Theta}}, \hat{\boldsymbol{\theta}}) = \hat{\boldsymbol{\theta}} + \hat{\boldsymbol{\Theta}}[\boldsymbol{I}_d + t\hat{\boldsymbol{\Theta}}]^{-1}(X_t - t\hat{\boldsymbol{\theta}}),$$

*and the rotating vector field:*

$$\hat{v}_{rOT}(X_t, t, \hat{\boldsymbol{\Theta}}^{\boldsymbol{R}}_{rOT}, \hat{\boldsymbol{\theta}}) = \hat{\boldsymbol{\theta}} + \hat{\boldsymbol{\Theta}}^{\boldsymbol{R}}_{rOT}[\boldsymbol{I}_d + t\hat{\boldsymbol{\Theta}}^{\boldsymbol{R}}_{rOT}]^{-1}(X_t - t\hat{\boldsymbol{\theta}}),$$

*where $X_t = (1-t)X_0 + tX_1$, $\hat{\boldsymbol{\Theta}}^{\boldsymbol{R}} = \boldsymbol{M}_d^{1/2} - \boldsymbol{I}_d$, $\hat{\boldsymbol{\Theta}}_{rOT} = \boldsymbol{M}_d^{1/2}\boldsymbol{R} - \boldsymbol{I}_d$, $\hat{\boldsymbol{\theta}} = \mu$, and $\boldsymbol{R}$ is a rotation matrix with $\boldsymbol{R} \neq -I$. Furthermore, the function $\hat{v}_{OT}$ cannot be exactly represented by an MLP, CNN, Transformer architecture when given concatenated inputs $[X_t, t]$.*

An intuitive reason why this vector field cannot be represented with zero error by typical neural network parameterizations is the difficulty in capturing the matrix inverse term $[\boldsymbol{I}_d + t\hat{\boldsymbol{\Theta}}]^{-1}$. For experiments quantifying the training impact of this approximation error, see App. G.2 for full results.

**Proposition 1** (Informal). *Let $\pi_0 \sim \mathcal{N}(0, \boldsymbol{I}_d)$, and $\pi_1 \sim \mathcal{N}(\mu, \boldsymbol{M}_d)$, where $\boldsymbol{M}_d$ is a positive definite and symmetric matrix. Let the following push forward maps: $T_{OT}(X_0) = \boldsymbol{M}_d^{1/2}X_0 + \mu$, $T^{\boldsymbol{R}}_{rOT}(X_0) = \boldsymbol{M}_d^{1/2}\boldsymbol{R}X_0 + \mu$, and $T_{rand}(X_0) = X_1$, where $R$ is a rotation matrix. Let the following parameterisation be $v(X_t, t) = \boldsymbol{\theta} + \boldsymbol{\Theta}[I_d + t\boldsymbol{\Theta}]^{-1}(X_t - t\boldsymbol{\theta})$, where $X_t = (1-t)X_0 + tX_1$. Let:*

$$L_{MC}(\boldsymbol{\Theta}, \boldsymbol{\theta}, T, v) = \frac{1}{N}\sum_{s=1}^{N}\|T(X_0^{(s)}) - X_0^{(s)} - v(X_t^{(s)}, t, \boldsymbol{\Theta}, \boldsymbol{\theta})\|^2. \qquad (9)$$

*$(\hat{\Theta}, \hat{\theta})$ with $T_{\mathrm{OT}}$ and $(\hat{\Theta}_{\mathrm{rOT}}, \hat{\theta})$ with $T^R_{\mathrm{rOT}}$ achieve zero loss and zero gradient variance. If the rotation angle is not $180°$, trajectories remain straight, but any learned field that does not correspond to the true angle exhibits nonzero gradient variance. In addition, for $T_{\mathrm{rOT}}$ and $T_{\mathrm{rand}}$ evaluated with the $v_{\mathrm{OT}}$ class, gradient variance does not increase merely because multiple straight interpolants pass near one another (for analytical values check Appendix B.).*

A common misconception in flow matching is that gradient variance primarily originates at points $(x_t, t)$ where straight-line interpolants intersect [Gagneux et al., 2025, Fjelde et al., 2024]; Proposition 1 challenges this by showing that variance can be nonzero even for straight couplings with

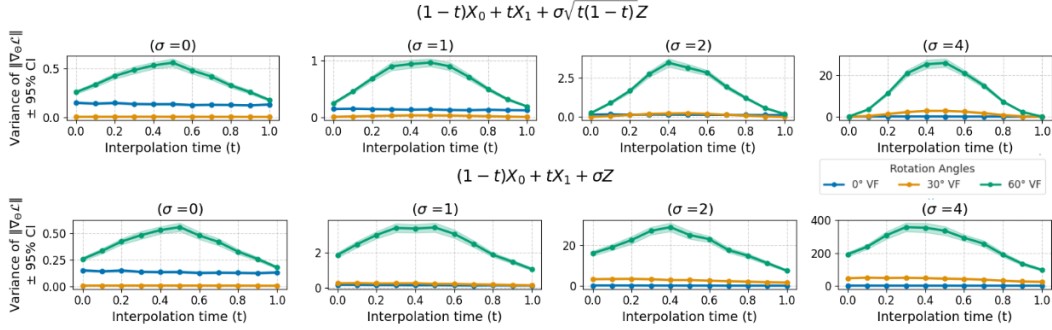

Figure 4: Gradient variance (with 95% confidence intervals) for Gaussian transport paired $(X_0, R_{30°}X_0 + \mu)$ under different noise levels ($\sigma$ of the two interpolants mentioned in the title of the figures). We compare vector fields inducing $0°$ (OT, blue), $30°$ (pair-optimal, orange), and $60°$ (non-OT, non-pair-optimal, green) rotations. Shaded regions show confidence intervals calculated over 100 bootstrap samples. As noise increases ($\sigma = 0 \rightarrow 4$), the optimal transport (OT) field exhibits significantly reduced variance ($p < 0.01$, paired t-test) while maintaining lower transport error.

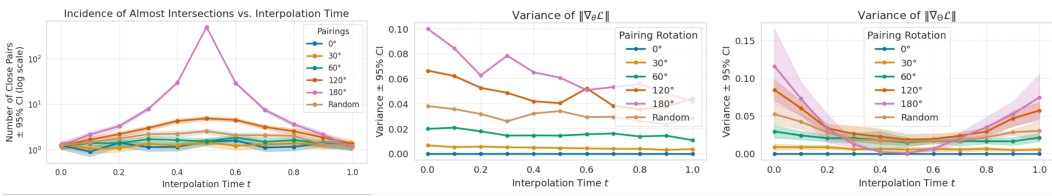

Figure 5: Gradient variance over time for several pairing types: $\left(X_0, \boldsymbol{M}_d^{1/2}\boldsymbol{R}_{k\circ}X_0 + \mu\right)$ with $k \in \{0, 30, 60, 120, 180\}$, and $(X_0, X_1)$ with $X_1 \sim \mathcal{N}(\mu, \boldsymbol{M}_d)$, over the OT vector field. Shaded regions indicate 95% confidence intervals computed via 100 bootstrap samples. Notably, variance does not increase in regions where interpolant trajectories come closer together: for example, with $180°$ rotation, all interpolants intersect at $t = 1/2$, yet variance is lowest there. Random pairings exhibit lower variance under the OT field than structured pairings with $120°$ (straight) or $180°$ (not-straight) rotations, highlighting that variance is not simply a function of interpolant density.

no intersections, while intersections or regions of high interpolant density do not, by themselves, induce elevated variance. Consequently, gradient variance under $L_{\mathrm{MC}}$ is not governed by geometric proximity of trajectories but by mismatch between the learned vector field and the pairing structure (e.g., a misaligned rotation). This also clarifies that low gradient variance does not certify transport optimality: when $v$ and $T$ are aligned (e.g., they apply the same rotation), both loss and gradients vanish, whereas under the OT field, even straight pairings can exhibit nonzero gradient variance (see Figures 4, 3, 6).

## 3.1 Empirical validation

This subsection studies empirically what happens with the gradient variance across choices of pairings, interpolants, and vector field classes.

**Key experiment:** Figure 6 illustrates the key experiment that motivates Section 4. We consider a deterministic pairing with $X_0 \sim \mathcal{N}(0, I_2)$ and $X_1 = R_{180°} X_0 + [5, 5]^\top$, so that all straight-line interpolants meet at the midpoint when $t = \frac{1}{2}$. Under noiseless interpolants, the learned vector field memorizes this ill-defined pairing and, at inference, reproduces the same mapping that rotates the source Gaussian by $180°$ and translates it by $[5, 5]^\top$. This occurs because the probability of sampling exactly $t = \frac{1}{2}$ during training is zero, so the vector field is never constrained at the intersection point; at inference, deterministic numerical integration effectively bypasses this singular location, yielding the same pairing. This provides a concrete failure case in which rectification does not resolve interpolant intersections but instead preserves the training-specific pairing.

**Under noiseless interpolants.** We now characterize the variance of the gradients over the noiseless interpolant $x_t = (1 - t)x_0 + tx_1$, as this is the commonly used type of interpolant in ReFlow architectures. Notably, when the vector field is OT, straight pairings can exhibit higher gradient variance than random pairings (see Figure 5). Conversely, with a non-OT vector field, the loss can display substantial variance for pairings that are themselves OT (see Figure 3). Furthermore, for

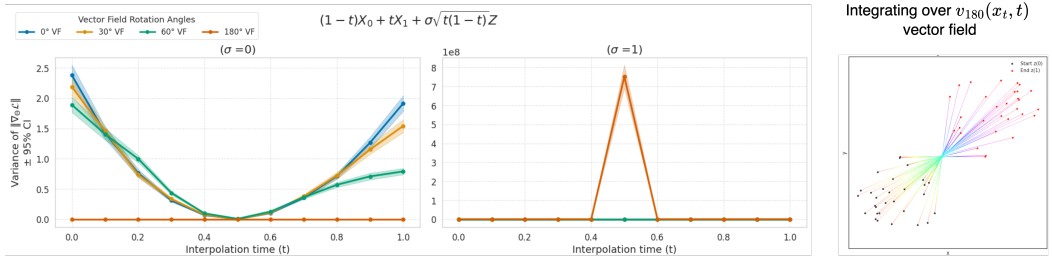

Figure 6: Integration over a vector field performing a $180°$ rotation. Although all interpolation paths intersect at $t = 1/2$, and $\mathbb{E}[X_1 - X_0 \mid X_t = x_{1/2}] \neq X_1 - X_0$, we still recover $180°$-rotated couplings. This is because the numerical integration procedure discretizes time and does not evaluate the vector field precisely at the point of intersection, effectively bypassing it.

straight non-OT pairings, the most stable solution is the corresponding pair-optimal (non-OT) vector field, Figure 6. Variance does not arise from intersections or regions where the interpolating lines come close together, and random pairings exhibit lower variance than structured couplings, such as a $120°$ rotation (see Figure 5).

**Variance under stochastic interpolants.** We now consider a noisy interpolant $x_t = (1 - t) x_0 + t x_1 + f(t, \sigma) Z, \quad Z \sim \mathcal{N}(0, \boldsymbol{I}_d)$, where $f : [0, 1] \to \mathbb{R}_{\geq 0}$ modulates the noise level over time (e.g., $f(t, \sigma) = \sigma \sqrt{t(1 - t)}$). As shown in Figure 4, and Figure 6, adding noise to the interpolant causes the optimal vector field to exhibit lower gradient variance than the pairing-optimal vector field. Furthermore, in the second frame of Figure 6, we can now see that at $t = 1/2$ the variance of the gradients is very high, meaning that, even though we never sample $t = 1/2$, the loss landscape has very high variance around that area for this vector field solution.

# 4 Rectified Flow Dynamics: Minimizer Memorizes

This section extends the experiment in Figure 6 by formalizing it through Proposition 2, which analyzes stagnation for finite datasets under deterministic training and clarifies its implications. We then validate these implications empirically on synthetic data and CelebA in Sections 4.2 and 4.3.

## 4.1 Theoretical Results

We now formalize two key properties of ReFlow: first, its tendency to stagnate once it reaches a straight coupling (Lemma 2), a known result previously established in Bansal et al. [2024], Liu [2022]; and second, its ability to memorize arbitrary deterministic pairings when trained on a finite dataset (Proposition 2), which represents the main novel contribution of this work.

**Lemma 2** (Idempotence of Rectified Flows). *Let $\pi_0, \pi_1$ be distributions on $\mathbb{R}^D$ with densities, and $(Z_0, Z_1)$ their straight-line coupling via linear interpolant $Z_t = (1 - t)Z_0 + tZ_1$. Subsequent Rectified Flow iterations with this noiseless interpolant yield identical couplings:*

$$\texttt{ReFlow}^{(k)}(Z_0, Z_1) = (Z_0, Z_1) \quad \forall k \geq 1.$$

This stagnation stems from the bijective relationship between interpolants $(Z_t, t)$ and initial pairs $(Z_0, Z_1)$, given by our assumption of the straightness of the deterministic couplings.

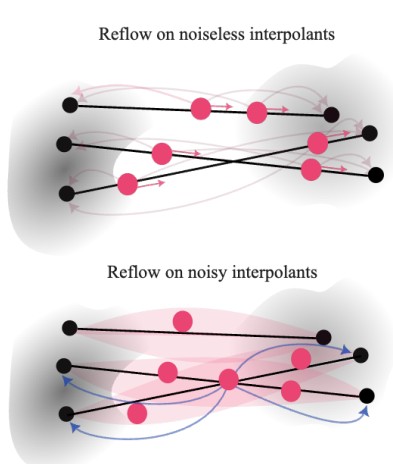

Reflow on noiseless interpolants

Reflow on noisy interpolants

Figure 7: Schematic showing the two types of interpolants. When training with deterministic pairings, sampling at an intersection has zero probability mass. As a result, the mapping $(Z_t, t) \mapsto (Z_0, T(Z_0), t)$ becomes injective, making it straightforward to define a vector field at these points by $v(Z_t, t) = Z_0 - T(Z_0)$. In contrast, for noisy interpolants, this injectivity no longer holds.

Some recent works have suggested that 2-ReFlow might be sufficient to recover straight pairings Lee et al. [2024], and Bansal et al. [2024] showed that 2 rectifications are enough under some regularity assumptions to straighten paths for isotropic Gaussian distributions. In the spirit of Figure 6 (see Appendix A), we provide an example where this is not possible for non-independent pairings.

**Counter Example 1** (1-ReFlow May Fail Under Mild Assumptions). *Even if the learned transport map $T(x_0)$ is injective (e.g., under standard Lipschitz and growth conditions; see Appendix A), the straight-line interpolant $I(x_0, T(x_0), t)$ need not be. For instance, a rotation-based map like $T(x_0) = R_{180°} x_0 + 5$ (which can be realized by a continuous vector field) leads to overlapping interpolants, breaking injectivity of $(x_t, t) \mapsto x_0$. As a result, a new ReFlow step cannot reconstruct $(x_0, x_1)$ and fails to straighten the coupling. See Appendix D for details and a visual example.*

Relaxing the straightness assumption to consider general deterministic couplings and finite training datasets leads to a nuanced but equally critical limitation:

**Proposition 2** (The Minimizer Memorizes). *Let $\pi_0$ and $\pi_1$ be two probability densities on $\mathbb{R}^D$, and let $T : \mathbb{R}^D \rightarrow \mathbb{R}^D$ be a deterministic transport map pushing $\pi_0$ to $\pi_1$ (so if $Z_0 \sim \pi_0$, then $Z_1 = T(Z_0) \sim \pi_1$). Suppose we draw a finite dataset of $N$ i.i.d. samples $\{Z_0^{(i)}, Z_1^{(i)} = T(Z_0^{(i)})\}_{i=1}^N$, and for each $i$ we also sample $m$ time-points $\{t^{(i,j)}\}_{j=1}^m \subset [0,1]$ (e.g. uniformly). Let $Z_t^{(i,j)} = (1 - t^{(i,j)}) Z_0^{(i)} + t^{(i,j)} Z_1^{(i)}$. Define the empirical loss over this doubly indexed dataset by*

$$L_{\mathrm{MC}}^{\mathrm{det}}(v_\Theta) = \frac{1}{N\,m} \sum_{i=1}^N \sum_{j=1}^m \left\| \left(Z_1^{(i)} - Z_0^{(i)}\right) - v_\Theta\left(Z_t^{(i,j)}, t^{(i,j)}\right) \right\|^2.$$

*Then there exists a (deterministic) vector field $v$ attaining zero loss: $L_{\mathrm{MC}}^{\mathrm{det}}(v) = 0$.*

Next, we discuss the implications of Lemma 2 and Proposition 2.

**Remark 1** (Condition for Recovering the Same Couplings). *Defining a look-up vector field over the training data—or approximating it via the Universal Approximation Theorem (UAT)—does not guarantee that integration will recover the original pairings. To ensure this, the integration process must traverse the specific time steps $\{t^{(i,j)}\}_{j=1}^m$ associated with each training pair $X_{(i)}$. These are the only points along the interpolation $x_t = (1 - t)x_0 + tx_1$ where we are guaranteed that $v(x_t, t) = x_1 - x_0$. While this condition might appear restrictive, the proposition remains valid for any natural number $m$.*

This insight arose from empirical observations. For example, when training a neural network on pairs $(X_0, T(X_0))$ with $T(X_0) = R_{180°} X_0 + \mu$, we found that—despite all interpolants intersecting at $t = 1/2$—the model successfully learned the rotation and preserved the pairings during integration. See Figure 6 for a visualization.

**Remark 2** (Noise Breaks Proof Assumptions). *A key advantage of using noisy interpolants is that they break the bijection between $(x_t, t)$ and $(x_0, T(x_0))$, thereby violating the key steps of the proofs of Lemma 2 and Proposition 2. This disruption prevents the model from becoming stuck in suboptimal or deterministic straight pairings. As shown in Figure 4, this can lead to learning vector fields that are closer to the optimal solution.*

*Why Doesn't CFM Memorize Random Pairings?* Consider $x_1$ a finite dataset of targets, while at each training step, the source $x_0$ is independently drawn from a continuous distribution. Initially, we hypothesized that CFM could memorize these random couplings, since Proposition 3 (Appendix C) shows that interpolants intersect at $x_t$ with probability zero. However, as noted in Remark 2, the independent sampling of $x_0$ from continuous distributions breaks the bijection between $(x_t, t)$ and $(x_0, x_1)$.

While Rectified Flows can help straighten trajectories, they risk collapsing onto memorized pairings at the first iteration, when deterministic couplings are formed. This memorization undermines generalization and highlights a limitation of relying solely on early deterministic transport.

## 4.2 Memorization for Mixture of Gaussian

To probe CFM's tendency to memorize deterministic pairings, we train two variants—one using noiseless interpolants and one with added noise—on an CFM model that maps $\pi_0 = \mathcal{N}(0, \boldsymbol{I}_d)$ to a target Gaussian mixture $\pi_1$ (more details for the experiments in Appendix F). We evaluate in both low ($d = 3$) and high ($d = 50$) dimensions using three metrics (log-likelihood under the true mixture, MMD, and Sinkhorn distance) and four comparisons: (i) *Gen*: generated held-out vs. true samples, (ii) *Mem*: generated vs. training pair integrations, (iii) *True*: true vs. true reference, and (iv) *Data*: training pair vs. true samples. This setup isolates whether noiseless CFM simply memorizes its finite dataset, and whether stochastic interpolants improve generalization without sacrificing sample quality.

As shown in Table 1, CFM tends to memorize the deterministic pairings it is trained on, reflected in very low memorization distances but weaker generalization. In contrast, CFM($\sigma = 0.05$) demonstrates consistently lower distances to the true distribution across both dimensions, suggesting superior generalization and less reliance on memorized pairings.

## 4.3 Memorization for CelebA

To empirically validate Proposition 2, we conduct experiments on the CelebA dataset using Conditional Flow Matching (CFM), with ground-truth optimal transport (OT) pairings from Korotin et al. [2021] as reference. Our primary objective is to test the claim that CFM trained with deterministic interpolants memorizes its input pairings, as predicted by theory.

We compare two versions of CFM: CFM($\sigma = 0$): trained using deterministic (noiseless) interpolants, CFM($\sigma = 0.05$): trained using slightly noisy interpolants.

Both models use identical U-Net architectures. The noise parameter $\sigma$ is not chosen for improved performance, but specifically to break the injectivity between data and interpolant trajectories—thereby disabling the memorization pathway outlined in Proposition 2.

**Adversarial Pairings.** To directly probe memorization, we construct an adversarial dataset by shuffling the targets of the Korotin et al. [2021] optimal OT pairings. This breaks correct transport structure, producing random pairings with no coherent OT semantics. The model variants are compared on two criteria: generalization (L2 error to true OT targets), and Memorization (L2 error to the shuffled (training) targets). We measure both on held-out subsets (5K and 50K samples). Lower L2 to true OT indicates better generalization, while low L2 to shuffled indicates greater memorization.

**Simulated 1-ReFlow.** We simulate a 1-step ReFlow scenario to test if iterative flow models reinforce memorization further—a central prediction of Proposition 2. Here, a base CFM model is first trained on random (non-OT) pairings, then used to generate endpoints via ODE integration. We then retrain CFM on these generated endpoints.

Memorization effects diminish with larger datasets, as fixed model capacity makes perfect memorization infeasible for 50K samples compared to 5K. However, optimization still favors memorization when possible, as predicted by Proposition 2. Larger datasets impede perfect memorization, but do not alter the underlying objective.

Our experiments provide direct empirical support for Proposition 2: CFM trained on deterministic interpolants reliably memorizes its inputs, even when those pairings are random or flawed. Introducing noise to interpolants breaks the pathway for memorization, enabling principled generalization toward the true OT map. These findings highlight a key limitation of deterministic training regimes for flow-based generative models on empirical datasets. For completion we mention FID values of these models in Appendix H.

## 5 Conclusion

We analyzed flow-based models through the lens of gradient variance and showed that deterministic interpolants—especially in Rectified Flows—can lead to memorization and stagnation. Noise, whether explicit (stochastic interpolants) or implicit (random pairings), breaks this effect, improving both generalization and sample quality. Our theoretical results and experiments on Gaussians,

Table 1: Comparison of CFM and CFM with stochastic interpolants (CFM($\sigma = 0.05$)) across low and high dimensions. CFM($\sigma = 0.05$) consistently generates samples that better match the target distribution (as indicated by lower MMD and Sinkhorn distances), without resorting to memorization, as evidenced by its strong performance on both generated and true data metrics. This highlights CFM($\sigma = 0.05$)'s superior generalization and sample quality compared to CFM. Experiments done over 10 seeds, the mean is presented, for full table check Appendix F.

| Dimension | | 3 | | | | 50 | | | |
|---|---|---|---|---|---|---|---|---|---|
| | | Gen | Mem | True | Data | Gen | Mem | True | Data |
| CFM ($\sigma = 0$) | LogProb | 4.0150 | 4.0890 | 4.1330 | 4.0155 | 54.8299 | 53.6502 | 52.5094 | 53.6244 |
| | MMD | 0.0034 | $1.758 \times 10^{-6}$ | 0.0014 | 0.0032 | 0.0021 | $9.089 \times 10^{-6}$ | 0.0020 | 0.0019 |
| | Sinkhorn | 0.0730 | $1.411 \times 10^{-5}$ | 0.0637 | 0.0790 | 15.1900 | 0.0045 | 14.3221 | 15.7400 |
| CFM ($\sigma = 0.05$) | LogProb | 4.1270 | 4.0960 | 4.1330 | 4.0155 | 54.7220 | 53.8890 | 52.5094 | 53.6244 |
| | MMD | 0.0018 | $3.105 \times 10^{-5}$ | 0.0014 | 0.0032 | 0.0020 | $6.09 \times 10^{-5}$ | 0.0020 | 0.0019 |
| | Sinkhorn | 0.0680 | $3.557 \times 10^{-4}$ | 0.0637 | 0.0790 | 15.1689 | 0.0304 | 14.3221 | 15.7400 |

Table 2: Adversarial Pairings Results: These results show that CFM with deterministic interpolants ($\sigma = 0$) strongly memorizes non-optimal shuffled targets, minimizing training loss even on arbitrary pairings. Adding slight noise ($\sigma = 0.05$) disrupts this, enabling the model to resist overfitting and generalize toward the true OT map.

| Metric | CFM($\sigma = 0.05$) | CFM($\sigma = 0$) |
|---|---|---|
| 5K Generalization (L2 to OT) | $34.25 \pm 7.54$ | $50.40 \pm 16.73$ |
| 5K Memorization (L2 to Shuffled) | $55.02 \pm 16.51$ | $28.57 \pm 5.49$ |
| 50K Generalization (L2 to OT) | $30.05 \pm 6.77$ | $46.78 \pm 14.87$ |
| 50K Memorization (L2 to Shuffled) | $56.48 \pm 18.55$ | $45.98 \pm 11.85$ |

Table 3: Simulated 1-ReFlow: The findings are consistent: deterministic interpolants ($\sigma = 0$) facilitate memorization of training pairings, while a small amount of noise substantially increases resistance to memorization and restores generalization ability.

| Metric | CFM($\sigma = 0.05$) | CFM($\sigma = 0$) |
|---|---|---|
| 5K Generalization (L2 to OT) | $31.35 \pm 7.38$ | $43.35 \pm 14.21$ |
| 5K Memorization (L2 to Generated) | $25.08 \pm 8.59$ | $8.63 \pm 1.76$ |
| 50K Generalization (L2 to OT) | $32.54 \pm 8.86$ | $38.16 \pm 11.37$ |
| 50K Memorization (L2 to Generated) | $16.29 \pm 4.82$ | $12.76 \pm 4.01$ |

Gaussian Mixtures and CelebA demonstrate that variance-sensitive optimization prefers suboptimal flows unless stochasticity is introduced. These findings challenge the reliability of rectified flows and underscore the importance of noise in guiding models toward better transport.

**Limitations.** Our evaluation focuses on: Gaussians, Gaussian mixtures, and CelebA as the sole real-world benchmark; broader modalities and higher resolutions may surface additional behaviors. While we critique Rectified Flows' propensity to memorize deterministic pairings, we do not propose a new remedy beyond introducing stochasticity into interpolants. Notably, adding noise is known to steer training toward entropic OT [De Bortoli et al., 2024] over many iterations, which in practice manifests as improved generalization rather than a principled cure for deterministic memorization.

# 6 Acknowledgements

TR and MB are supported by EPSRC Turing AI World-Leading Research Fellowship No. EP/X040062/1 and EPSRC AI Hub on Mathematical Foundations of Intelligence: An "Erlangen Programme" for AI No. EP/Y028872/1. SD is supported by the Clarendon Scholarship and NERC DTP studentship (NE/S007474/1). FV declares no specific funding for this work.

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

# A   Assumptions

**Assumption 1.** *Suppose that $v(x, t)$ satisfies the following conditions:*

- **Lipschitz continuity in** $x$**:** *There exists $L > 0$ such that for all $x, y \in \mathbb{R}^D$ and $t \in [0, 1]$,*

$$\|v(x, t) - v(y, t)\| \leq L\|x - y\|. \tag{10}$$

- **Linear growth:** *There exists $C > 0$ such that for all $x \in \mathbb{R}^D$ and $t \in [0, 1]$,*

$$\|v(x, t)\| \leq C(1 + \|x\|). \tag{11}$$

With Assumption 1 in place we can conclude via Arnold [1992], that the ODE $dx_t = v(x_t, t)dt$ admits unique solutions for all initial conditions $x_0$, and so that $T(x_0) = x_0 + \int_0^1 v(x_t, t)dt$ is an injective map.

# B   Proofs - Gaussian Setting:

**Extra-Lemma 1.** *Let $q(x) = \frac{1}{x}$, with $x \in \mathbb{R}^d$. There is no finite parameterization of a Multilayer Perceptron (MLP) that can represent this function with zero error.*

*Proof.* Assume, for the sake of contradiction, that there exists a finite MLP that represents $q(x)$ exactly. To avoid issues at $x = 0$ where $q(x)$ becomes undefined, we restrict the domain of $x$ to $\mathbb{R} \setminus [a, b]$, where $0 \notin [a, b]$ and $a, b \in \mathbb{R}$. This ensures that $q(x)$ is Lipschitz continuous on this domain.

Next, consider the scaling property of $q(x)$: for any scalar $c \neq 0$, $q(cx) = \frac{1}{c}q(x)$. This property implies that $q(x)$ scales inversely with $c$.

However, standard MLPs do not inherently possess this scaling property. For an MLP to exhibit this behavior, its activation functions would need to satisfy $\sigma(cx) = \frac{1}{c}\sigma(x)$, which is not true for commonly used activation functions such as ReLU or tanh.

Since the desired function $q(x)$ exhibits a scaling property that standard MLPs cannot replicate, we conclude that no finite MLP can represent $q(x)$ exactly. $\qquad\square$

**Lemma 1.** *Let $X_0 \sim \mathcal{N}(0, \boldsymbol{I}_d)$ and $X_1 \sim \mathcal{N}(\mu, \boldsymbol{M}_d)$, where $\boldsymbol{M}_d$ is a positive definite and symmetric matrix. The OT vector field is given by*

$$\hat{v}_{OT}(X_t, t, \hat{\boldsymbol{\Theta}}, \hat{\boldsymbol{\theta}}) = \hat{\boldsymbol{\theta}} + \hat{\boldsymbol{\Theta}}[\boldsymbol{I}_d + t\hat{\boldsymbol{\Theta}}]^{-1}(X_t - t\hat{\boldsymbol{\theta}}),$$

*and the rotating vector field:*

$$\hat{v}_{rOT}(X_t, t, \hat{\boldsymbol{\Theta}}_{rOT}, \hat{\boldsymbol{\theta}}) = \hat{\boldsymbol{\theta}} + \hat{\boldsymbol{\Theta}}_{rOT}[\boldsymbol{I}_d + t\hat{\boldsymbol{\Theta}}_{rOT}]^{-1}(X_t - t\hat{\boldsymbol{\theta}}),$$

*where $X_t = (1 - t)X_0 + tX_1$, $\hat{\boldsymbol{\Theta}} = \boldsymbol{M}_d^{1/2} - \boldsymbol{I}_d$, $\hat{\boldsymbol{\Theta}}_{rOT} = \boldsymbol{M}_d^{1/2}\boldsymbol{R}_{k\circ} - \boldsymbol{I}_d$, $\hat{\boldsymbol{\theta}} = \mu$, and $\boldsymbol{R}_{k\circ}$ is a rotation matrix with $\boldsymbol{R}_{k\circ} \neq -\boldsymbol{R}_{k\circ}$. Furthermore, the function $\hat{v}_{OT}$ cannot be exactly represented by an MLP, CNN, Transformer architecture when given concatenated inputs $[X_t, t]$.*

*Proof.* The optimal transport map from $\mathcal{N}(0, \boldsymbol{I}_d)$ to $\mathcal{N}(\mu, M_d)$ is $T_{OT}(x) = \mu + M_d^{1/2}x$. Thus, if we denote the optimal coupling by $X_0^*$ and $X_1^*$, we have $X_1^* = T(X_0^*) = \mu + M_d^{1/2}X_0^*$. The displacement interpolation is defined as $X_t = (1 - t)X_0^* + tX_1^*$. Substitute the expression for $X_1^*$ into the interpolation:

$$X_t = (1 - t)X_0^* + t\left(\mu + M_d^{1/2}X_0^*\right) = \left[(1 - t)\boldsymbol{I}_d + t\,M_d^{1/2}\right]X_0^* + t\mu. \tag{12}$$

Solving for $X_0^*$ gives

$$X_0^* = \left[(1 - t)\boldsymbol{I}_d + t\,M_d^{1/2}\right]^{-1}(X_t - t\mu).$$

Now, substitute $X_0^*$ back into the transport map to get

$$X_1^* = \mu + M_d^{1/2} X_0^* = \mu + M_d^{1/2}\Big[(1-t)\boldsymbol{I}_d + t\, M_d^{1/2}\Big]^{-1}(X_t - t\mu). \tag{13}$$

The optimal displacement (vector field) is defined as $v(X_t, t) = X_1^* - X_0^*$. Therefore, we have

$$v_{OT}(X_t, t) = \mu + M_d^{1/2}\Big[(1-t)\boldsymbol{I}_d + t\, M_d^{1/2}\Big]^{-1}(X_t - t\mu) \tag{14}$$

$$- \Big[(1-t)\boldsymbol{I}_d + t\, M_d^{1/2}\Big]^{-1}(X_t - t\mu) \tag{15}$$

$$= \mu + \Big(M_d^{1/2} - \boldsymbol{I}_d\Big)\Big[(1-t)\boldsymbol{I}_d + t\, M_d^{1/2}\Big]^{-1}(X_t - t\mu). \tag{16}$$

This is the desired expression for the vector field.

For $v_{rOT}$, the computation is analogous, with the key difference that the transport map is now given by $T_{rOT}(x) = \mu + M_d^{1/2}RX_0$. This defines a valid pushforward map, since for any rotation matrix $R_{k\circ}$, if $Z \sim \mathcal{N}(0, \boldsymbol{I}_d)$, then $RZ \sim \mathcal{N}(0, \boldsymbol{I}_d)$ as well, due to the rotational invariance of the standard Gaussian.

Proceeding as before, we solve for $X_0^*$, substitute it back into the transport map, and derive the corresponding vector field. This yields the expression for $v_{rOT}$. The derivation follows the same steps as in the case of $v_{OT}$, with the only difference being that $M_d^{1/2}$ is replaced by $M_d^{1/2}R$.

Also we can't have $R = -R$ (180° rotation), because then the inverse $\Big[(1-t)\boldsymbol{I}_d - t\boldsymbol{M}_d^{1/2}\Big]^{-1} = \Big[I_d - t(\boldsymbol{M}_d^{1/2} + \boldsymbol{I})\Big]^{-1}$ is not computable when $t(\boldsymbol{M}_d^{1/2} + \boldsymbol{I}) = \boldsymbol{I}$, (for example for $\boldsymbol{M}_d = \boldsymbol{I}$, and $t = 1/2$). For a more complete approach, see Liu [2022].

We proceed to prove that this parameterization cannot be represented with zero error by an MLP. Representing the function $\frac{\theta}{1+t\theta}$ with zero error is equivalent to representing $\frac{1}{\theta}$ with zero error. By applying Extra-Lemma 1, we conclude that such a representation is not possible. $\square$

**Proposition 1.** *Let $\pi_0 \sim \mathcal{N}(0, \boldsymbol{I}_d)$, and $\pi_1 \sim \mathcal{N}(\mu, \boldsymbol{M}_d)$, where $\boldsymbol{M}_d$ is a positive definite and symmetric matrix. Let the following push forward maps: $T_{OT}(X_0) = \boldsymbol{M}_d^{1/2}X_0 + \mu$, $T_{rOT}(X_0) = \boldsymbol{M}_d^{1/2}\boldsymbol{R}X_0 + \mu$, and $T_{rand}(X_0) = X_1$, where $R$ is a rotation matrix. Let the linear transformation be $v(X_t, t) = \boldsymbol{\theta} + \boldsymbol{\Theta}[I_d + t\boldsymbol{\Theta}]^{-1}(X_t - t\boldsymbol{\theta})$, where $X_t = (1-t)X_0 + tX_1$. Let:*

$$L_{MC}(\boldsymbol{\Theta}, \boldsymbol{\theta}, T, v) = \frac{1}{N}\sum_{s=1}^{N}\|T(X_0^{(s)}) - X_0^{(s)} - v(X_t^{(s)}, t, \boldsymbol{\Theta}, \boldsymbol{\theta})\|^2. \tag{17}$$

*Then, for optimal, $(\hat{\boldsymbol{\Theta}}, \hat{\boldsymbol{\theta}})$ we have $Var[\nabla_{\boldsymbol{\theta}}L_{MC}(\hat{\boldsymbol{\Theta}}, \hat{\boldsymbol{\theta}}, v_{OT}, T_{rOT})] = Var[\nabla_{\boldsymbol{\Theta}}L_{MC}(\hat{\boldsymbol{\Theta}}, \hat{\boldsymbol{\theta}}, v_{OT}, T_{OT})] = Var[\nabla_{\boldsymbol{\theta}}(\hat{\boldsymbol{\Theta}}, \hat{\boldsymbol{\theta}}, v_{rOT}, T_{rOT})] = Var[\nabla_{\boldsymbol{\Theta}}L_{MC}(\hat{\boldsymbol{\Theta}}, \hat{\boldsymbol{\theta}}, v_{rOT}, T_{rOT})] = 0$, and $Var[\nabla_{\boldsymbol{\theta}}L_{MC}(\hat{\boldsymbol{\Theta}}, \hat{\boldsymbol{\theta}}, v_{OT}, T_{rOT}^R)] = \frac{4}{N}\|M_d 1\|^2$, and $Var[\nabla_{\boldsymbol{\Theta}}L_{MC}(\hat{\boldsymbol{\Theta}}, \hat{\boldsymbol{\theta}}, v_{OT}, T_{rOT}^R)] = \frac{8}{N}tr((M_d^{1/2}(R - \boldsymbol{I}))^2)$. Moreover, for the maps $T_{rOT}$ and $T_{rand}$ and $v_{OT}$ vector field, the variance of the gradients of the optimal vector field does not increase in regions where the interpolant trajectories come closer together.*

*Proof.* We break this proof into the computation of variance over time and the value at time zero to simplify computations.

**1. Way 1: for $t = 0$ but closed form is nice:** We compute gradient variance in two ways, one relies on the fact that when $T$ is deterministic there is a direct bijection between $X_t$ and $X_0$, this of

course won't give us the values in terms of $t$, but the formulas are somehow nicer and easier to work with. We are basically looking at the variance for $(X_0, 0)$

$$\text{Var}\left[\nabla_{\boldsymbol{\theta}} \frac{1}{N} \sum \left\| M_d^{1/2} X_0^{(n)} - X_0^{(n)} + \mu - \boldsymbol{\Theta} X_0^{(n)} - \boldsymbol{\theta} \right\|^2\right] \tag{18}$$

$$= \text{Var}\left[-\frac{2}{N} \sum \mathbf{1}^\top \left(\mu + (M_d^{1/2} - \boldsymbol{\Theta} - I) X_0^{(n)} - \boldsymbol{\theta}\right)\right] \tag{19}$$

$$= \frac{4}{N} \text{Var}\left[\mathbf{1}^\top (M_d^{1/2} - \boldsymbol{\Theta} - I) X_0^{(n)}\right] \tag{20}$$

$$= \frac{4}{N} \mathbf{1}^\top (M_d^{1/2} - \boldsymbol{\Theta} - I)(M_d^{1/2} - \boldsymbol{\Theta} - I)^\top \mathbf{1}. \tag{21}$$

$$\text{Var}\left[\nabla_{\boldsymbol{\Theta}_i} \frac{1}{N} \sum \left\| M_d^{1/2} X_0^{(n)} - X_0^{(n)} - \mu - \boldsymbol{\Theta} X_0^{(n)} - \boldsymbol{\theta} \right\|^2\right] \tag{22}$$

$$= \text{Var}\left[-\frac{2}{N} \sum X_0^\top \left(\mu + (M_d^{1/2} - I_d - \boldsymbol{\Theta}) X_0^{(n)} - \boldsymbol{\theta}\right)\right]. \tag{23}$$

**Gradient variance for $T_{OT}$ and $v_{OT}$**   For the rotated case:

Note that our vector field is optimal when $\hat{\boldsymbol{\Theta}} = M_d^{1/2} - I_d$ and $\hat{\boldsymbol{\theta}} = \mu$. Then we have:

$$\text{Var}\left[\nabla_{\boldsymbol{\theta}} L_{MC}(v(\hat{\boldsymbol{\Theta}}, \hat{\boldsymbol{\theta}}, X_0))\right] = 0, \tag{24}$$

and

$$\text{Var}\left[\nabla_{\boldsymbol{\Theta}} L_{MC}^{OT}(v(\hat{\boldsymbol{\Theta}}, \hat{\boldsymbol{\theta}}, X_0))\right] = \frac{4}{N} \text{Var}[X_0^\top(\mu - \hat{\boldsymbol{\theta}})] = \frac{4\|\mu - \hat{\boldsymbol{\theta}}\|^2}{N} = 0. \tag{25}$$

**Gradient variance for $T_{rOT}$ and $v_{OT}$**   For the rotated case:

$$\text{Var}\left[\nabla_{\boldsymbol{\theta}} \frac{1}{N} \sum \left\| M_d^{1/2} R X_0^{(n)} - X_0^{(n)} - \mu - \boldsymbol{\Theta} X_0^{(n)} - \boldsymbol{\theta} \right\|^2\right] \tag{26}$$

$$= \text{Var}\left[-\frac{2}{N} \sum \mathbf{1}^\top \left(\mu + (M_d^{1/2} R - \boldsymbol{\Theta} - I) X_0^{(n)} - \boldsymbol{\theta}\right)\right] \tag{27}$$

$$= \frac{4}{N} \text{Var}\left[\mathbf{1}^\top (M_d^{1/2} R - \boldsymbol{\Theta} - I) X_0^{(n)}\right]. \tag{28}$$

At the optimum, where $\hat{\boldsymbol{\Theta}} = M_d^{1/2} - I_d$, we obtain:

$$\frac{4}{N} \text{Var}\left[\mathbf{1}^\top (M_d^{1/2} R - M_d^{1/2}) X_0\right] = \frac{4}{N} \mathbf{1}^\top (M_d^{1/2}(R - I))(M_d^{1/2}(R - I))^\top \mathbf{1} \tag{29}$$

$$= \frac{4}{N} \mathbf{1}^\top (M_d^{1/2}(R - I))(R - I)^\top M_d^{1/2\top} \mathbf{1} \tag{30}$$

$$= \frac{4}{N} \mathbf{1}^\top M_d^{1/2}(2 I_d - R^\top - R) M_d^{1/2\top} \mathbf{1}. \tag{31}$$

For the variance of the gradient w.r.t. $\boldsymbol{\Theta}$ at the optimum, we get:

$$\text{Var}\left[\nabla_{\boldsymbol{\Theta}} L_{MC}^{rOT}(\hat{\boldsymbol{\Theta}}, \hat{\boldsymbol{\theta}})\right] = \frac{4}{N} \text{Var}\left[X_0^\top M_d^{1/2}(R - I) X_0\right] \tag{32}$$

$$= \frac{8}{N} \text{tr}\left((M_d^{1/2}(R - I))^2\right). \tag{33}$$

The last equality follows from the known variance formula of a quadratic form for Gaussian random vectors, specifically when $X_0 \sim \mathcal{N}(0, I)$.

**Gradient variance for $T_{rOT}$ and $v_{rOT}$**   For the rotated case:

$$\text{Var}\left[\nabla_{\boldsymbol{\theta}}\frac{1}{N}\sum\left\|\boldsymbol{M}_d^{1/2}\boldsymbol{R}X_0^{(n)} - X_0^{(n)} - \mu - \boldsymbol{\Theta}X_0^{(n)} - \boldsymbol{\theta}\right\|^2\right] \tag{34}$$

$$= \text{Var}\left[-\frac{2}{N}\sum\mathbf{1}^\top\left(\mu + (\boldsymbol{M}_d^{1/2}\boldsymbol{R} - \boldsymbol{\Theta} - \boldsymbol{I})X_0^{(n)} - \boldsymbol{\theta}\right)\right] \tag{35}$$

$$\tag{36}$$

At the optimum, where $\hat{\boldsymbol{\Theta}} = \boldsymbol{M}_d^{1/2}\boldsymbol{R} - \boldsymbol{I}_d$, we obtain:

$$\frac{4}{N}\text{Var}\left[\mathbf{1}^\top(\boldsymbol{M}_d^{1/2}\boldsymbol{R} - \boldsymbol{M}_d^{1/2}\boldsymbol{R})X_0\right] = 0 \tag{37}$$

For the variance of the gradient w.r.t. $\boldsymbol{\Theta}$ at the optimum, we get:

$$\text{Var}\left[\nabla_{\boldsymbol{\Theta}}L_{MC}^{rOT}(\hat{\boldsymbol{\Theta}},\hat{\boldsymbol{\theta}})\right] = \frac{4}{N}\text{Var}\left[X_0^\top \boldsymbol{M}_d^{1/2}(\boldsymbol{R} - \boldsymbol{R})X_0\right] \tag{38}$$

$$= 0 \tag{39}$$

**Way 2 - (for any $t$, closed form is not nice): Gradient variance over time**   Before analyzing different kinds of pairings that we want to analyze, first, we want a true formulation of the gradients:

$$\text{Var}[\nabla_{\boldsymbol{\theta}}L_{\text{MC}}^{type}(\hat{\boldsymbol{\Theta}},\hat{\boldsymbol{\theta}})] \tag{40}$$

$$= \text{Var}\left[\nabla_{\boldsymbol{\theta}}\frac{1}{N}\sum[\||(T_{type}(X_0^{(n)}) - X_0^{(n)} - v_{\boldsymbol{\Theta},\boldsymbol{\theta}}(X_t^{(n)},t)||^2]\right] \tag{41}$$

$$= \text{Var}\left[-\frac{2}{N}\sum(T_{type}(X_0^{(n)}) - X_0^{(n)} - \boldsymbol{\theta} - \boldsymbol{\Theta}[I_d + t\boldsymbol{\Theta}]^{-1}(X_t^{(n)} - t\boldsymbol{\theta}))^\top(-\mathbf{1} - t\boldsymbol{\Theta}[I_d + t\boldsymbol{\Theta}]^{-1}\mathbf{1})\right] \tag{42}$$

$$= \text{Var}\left[-\frac{2}{N}\sum(T_{type}(X_0^{(n)}) - X_0^{(n)} - \boldsymbol{\theta} - \boldsymbol{\Theta}[I_d + t\boldsymbol{\Theta}]^{-1}(X_t^{(n)} - t\boldsymbol{\theta}))^\top([I_d + t\boldsymbol{\Theta}]^{-1}\mathbf{1})\right] \tag{43}$$

For now, let $\boldsymbol{C}_{t,\boldsymbol{\Theta}} = [I_d + t\boldsymbol{\Theta}]^{-1}$. We arrive at a more simplified:

$$\text{Var}[\nabla_{\boldsymbol{\theta}}L_{\text{MC}}^{type}(\hat{\boldsymbol{\Theta}},\hat{\boldsymbol{\theta}})] = \frac{4}{N}\text{Var}[(T_{type}(X_0) - X_0 - \boldsymbol{\theta} - \boldsymbol{\Theta}[I_d + t\boldsymbol{\Theta}]^{-1}(X_t - t\boldsymbol{\theta}))^\top \boldsymbol{C}_{t,\boldsymbol{\Theta}}\mathbf{1}] \tag{44}$$

$$= \frac{4}{N}\text{Var}[(T_{type}(X_0) - X_0 - \boldsymbol{\Theta}[I_d + t\boldsymbol{\Theta}]^{-1}X_t)^\top \boldsymbol{C}_{t,\boldsymbol{\Theta}}\mathbf{1}] \tag{45}$$

$$= (\boldsymbol{C}_{t,\boldsymbol{\Theta}}\mathbf{1})^T\frac{4}{N}\text{Var}[(T_{type}(X_0) - X_0 - \boldsymbol{\Theta}[I_d + t\boldsymbol{\Theta}]^{-1}X_t)^\top]\boldsymbol{C}_{t,\boldsymbol{\Theta}}\mathbf{1} \tag{46}$$

Moving forward to gradients with respect to $\boldsymbol{\Theta}$ we get:

$$\text{Var}[\nabla_{\boldsymbol{\Theta}}L_{\text{MC}}^{type}(\hat{\boldsymbol{\Theta}},\hat{\boldsymbol{\theta}})] \tag{47}$$

$$= \text{Var}\left[\nabla_{\boldsymbol{\Theta}}\frac{1}{N}\sum[\||(T_{type}(X_0^{(n)}) - X_0^{(n)} - v_{\boldsymbol{\Theta},\boldsymbol{\theta}}(X_t^{(n)},t^{(n)})||^2]\right] \tag{48}$$

$$= \frac{4}{N}\text{Var}\left[(T_{type}(X_0) - X_0 - \boldsymbol{\theta} - \boldsymbol{\Theta}[I_d + t\boldsymbol{\Theta}]^{-1}(X_t - t\boldsymbol{\theta}))^\top((t\boldsymbol{C}_{t,\boldsymbol{\Theta}} - I)\boldsymbol{C}_{t,\boldsymbol{\Theta}}(X_t - t\boldsymbol{\theta}))\right] \tag{49}$$

$$\tag{50}$$

**CASE 1 (OT):** This case is nice because we just need to prove that:

$$Var[(T_{OT}(X_0) - X_0 - \boldsymbol{\theta} - \boldsymbol{\Theta}[I_d + t\boldsymbol{\Theta}]^{-1}(X_t - t\boldsymbol{\theta}))^\top] = \mathbf{0}, \tag{51}$$

which actually happens once we replace $\boldsymbol{\Theta} = (M^{1/2} - I)$, and $\boldsymbol{\theta} = \mu$. We will show this for completion:

$$M^{1/2}X_0 - X_0 - \boldsymbol{\Theta}[I_d + t\boldsymbol{\Theta}]^{-1}X_t \tag{52}$$

$$= (M^{1/2} - I)(I_d + t(M^{1/2} - I))^{-1}(X_0 + t(M^{1/2} - I)X_0 - tM^{1/2}X_0 - (1-t)X_0) \tag{53}$$

$$= (M^{1/2} - I)(I_d + t(M^{1/2} - I))^{-1}\boldsymbol{0} = \boldsymbol{0} \tag{54}$$

**Remark** If our network is not quite at the optima so $M^{1/2}X_0 - X_0 - \boldsymbol{\Theta}[I_d + t\boldsymbol{\Theta}]^{-1}X_t = \epsilon$ we can notice that the variance of $\boldsymbol{\theta}$ and $\boldsymbol{\Theta}$ decrease in time because $\frac{1}{1+tx}$ and $\frac{1+tx+t}{(1+tx)^2}$ are decreasing in t.

**No connection between gradient variance and crossings**    To show the lack of connection between crossings and variance of gradients, it is enough to offer counter-examples. In the rotation case, we will look at the variance when the rotation matrix does a $180°$ degree rotation, which means all interpolation lines should meet at $t = 1/2$. As we will show, there is no peak in variance at time $t = 1/2$. In the Gaussian case, we will observe a very similar behaviour.

**Case 2 (rOT):**

*1. For $\boldsymbol{\theta}$ the variance of the gradients will be:*

$$\text{Var}[\nabla_{\boldsymbol{\theta}}L_{\text{MC}}^{type}(\hat{\boldsymbol{\Theta}}, \hat{\boldsymbol{\theta}})] = \frac{4}{N}\mathbf{1}^T C_{t,\hat{\boldsymbol{\Theta}}}^T Var[(M^{1/2}R - I - \hat{\boldsymbol{\Theta}}C_{t,\hat{\boldsymbol{\Theta}}}(t(M^{1/2}R - I) - I))X_0]C_{t,\hat{\boldsymbol{\Theta}}}\mathbf{1} \tag{55}$$

$$= \frac{4}{N}\mathbf{1}^T C_{t,\hat{\boldsymbol{\Theta}}}^T A^T A C_{t,\hat{\boldsymbol{\Theta}}}\mathbf{1}, \tag{56}$$

where $A = M^{1/2}R - I - (M^{1/2} - I)[I + t(M^{1/2} - I)]^{-1}(t(M^{1/2}R - I) + I)$, with $A^T A \approx M + I - R - R^T + 2(1-t)(R-I)^T(M^{1/2} - I)(R-I)$, for small $M^{1/2} - I$ (approximation obtained via taylor expansion).

**Counter-example** For $R = -I$ and $M^{1/2} = 2I$(which is $180°$ rotation so all interpolants meet at one point) we have $A = -3I - [I - tI]^{-1}(-3It + I)$ so $A^T A = (\frac{16}{1-t})^2 I$. The variance in this case will be increasing for $t \in [0, 1]$. This aligns with our empirical results; however, it does not align with what is believed in the literature, that the variance would peak at $t = 1/2$, because crossings increase variance.

*2. For $\boldsymbol{\Theta}$, the variance of the gradients will be:*

$$\text{Var}[\nabla_{\boldsymbol{\Theta}}L_{\text{MC}}^{rOT}(\hat{\boldsymbol{\Theta}}, \hat{\boldsymbol{\theta}})] \tag{57}$$

$$= \frac{4}{N}\text{Var}\left[(T_{rOT}(X_0) - X_0 - \boldsymbol{\theta} - \boldsymbol{\Theta}[I_d + t\boldsymbol{\Theta}]^{-1}(X_t - t\boldsymbol{\theta}))^\top((tC_{t,\boldsymbol{\Theta}} - I)C_{t,\boldsymbol{\Theta}}(X_t - t\boldsymbol{\theta}))\right] \tag{58}$$

$$= \frac{4}{N}\text{Var}[(M^{1/2}R - I - (M^{1/2} - I)[I_d + t(M^{1/2} - I)]^{-1}(tM^{1/2}R + (1-t)\boldsymbol{I})X_0)^T \tag{59}$$

$$(tC_{t,\boldsymbol{\Theta}} - I)C_{t,\boldsymbol{\Theta}}(tM^{1/2} + (1-t)I)X_0] \tag{60}$$

$$= \frac{4}{N}\text{Var}[(AX_0)^T(tC_{t,\boldsymbol{\Theta}} - I)C_{t,\boldsymbol{\Theta}}(tM^{1/2}R + (1-t)I)X_0] = \frac{4}{N}\text{Var}[X_0^T A^T B X_0] = 2Tr((AB)^2) \tag{61}$$

**Counter-example** Let $M^{1/2} = 2I$, and $R = -I$. Then $A = \frac{4}{1+t}I$, and $B = \frac{(3t-1)}{(1+t)^2}$ so $\text{Var}[\nabla_{\boldsymbol{\Theta}}L_{\text{MC}}^{rOT}(\hat{\boldsymbol{\Theta}}, \hat{\boldsymbol{\theta}})] = \frac{4}{N}Var[\frac{16}{(1+t)^2}\frac{(3t-1)}{(1+t)^2}X_0^T X_0] = \frac{8d}{N}\frac{(16(3t-1))^2}{(1+t)^8}$ which is decreasing for $t \in [1/9, 1/3]$ and increasing $t \in [1/3, 1]$.

**Case 3 (Random):**

*1. For $\boldsymbol{\theta}$ the variance of the gradients will be:*

$$\text{Var}[\nabla_{\boldsymbol{\theta}} L_{\text{MC}}^{type}(\hat{\boldsymbol{\Theta}}, \hat{\boldsymbol{\theta}})] = \frac{4}{N}\mathbf{1}^T \boldsymbol{C}_{t,\hat{\boldsymbol{\Theta}}}^T Var[(X_1 - X_0 - \hat{\boldsymbol{\Theta}}\boldsymbol{C}_{t,\hat{\boldsymbol{\Theta}}}(tX_1 - (1-t)X_0))]\boldsymbol{C}_{t,\hat{\boldsymbol{\Theta}}}\mathbf{1} \tag{62}$$

$$= \frac{4}{N}\mathbf{1}^T \boldsymbol{C}_{t,\hat{\boldsymbol{\Theta}}}^T [Var[(I - \hat{\boldsymbol{\Theta}}\boldsymbol{C}_{t,\hat{\boldsymbol{\Theta}}}t)X_1] + Var[(-I - \hat{\boldsymbol{\Theta}}\boldsymbol{C}_{t,\hat{\boldsymbol{\Theta}}}(1-t)\boldsymbol{I})X_0]]\boldsymbol{C}_{t,\hat{\boldsymbol{\Theta}}}\mathbf{1}, \tag{63}$$

because of independence of $X_0$, and $X_1$. Continuing:

$$Var[(I - \hat{\boldsymbol{\Theta}}\boldsymbol{C}_{t,\hat{\boldsymbol{\Theta}}}t)X_1] = (I - t(M^{1/2} - I)(I + t(M^{1/2} - I))^{-1})^T M \tag{64}$$

$$(I - t(M^{1/2} - I)(I + t(M^{1/2} - I))^{-1}) \tag{65}$$

$$= (I + t(M^{1/2} - I))^{-1}M(I + t(M^{1/2} - I))^{-1}, \tag{66}$$

and,

$$Var[(I - \hat{\boldsymbol{\Theta}}\boldsymbol{C}_{t,\hat{\boldsymbol{\Theta}}}(1-t))X_0] = (I - (1-t)(M^{1/2} - I)(I + t(M^{1/2} - I))^{-1})^T \tag{67}$$

$$(I - (1-t)(M^{1/2} - I)(I + t(M^{1/2} - I))^{-1}). \tag{68}$$

**Counter-example:** For $M^{1/2} = 2I$ we have $Var[(I - \hat{\boldsymbol{\Theta}}\boldsymbol{C}_{t,\hat{\boldsymbol{\Theta}}}t)X_1] = 4\frac{1}{(1+t)^2}I$ which is decreasing for $t \in [0,1]$. We have $Var[(I - \hat{\boldsymbol{\Theta}}\boldsymbol{C}_{t,\hat{\boldsymbol{\Theta}}}(1-t))X_0] = 4\frac{t^2}{(1+t)^2}I$. That is $4\frac{t^2+1}{(1+t)^2}$ which is decreasing for $t \in [0,1]$. We have the overall variance:

$$\text{Var}[\nabla_{\boldsymbol{\theta}} L_{\text{MC}}^{type}(\hat{\boldsymbol{\Theta}}, \hat{\boldsymbol{\theta}})] = 4\mathbf{1}^T \frac{1}{1+t}\frac{t^2+1}{(1+t)^2}\frac{1}{1+t}I\mathbf{1} = 4d\frac{t^2+1}{(1+t)^4}, \tag{69}$$

which is decreasing in $t \in [0,1]$.

## C   Proofs - ReFlow

**Lemma 2.** *Let $\pi_1$ and $\pi_2$ be two distributions on $\mathbb{R}^N$ admitting densities, and let $(Z_0, Z_1)$ be their straight-line coupling. If we apply Rectified Flows again using the noise-free interpolant, we recover the same coupling.*

*Proof.* Since $(Z_0, Z_1)$ is the straight coupling, for each $t \in [0,1]$ the interpolation

$$Z_t \;=\; (1-t)\,Z_0 \;+\; t\,Z_1$$

is $\sigma(Z_0, Z_1)$–measurable. Define

$$v(z,t) \;:=\; \mathbb{E}\big[\,Z_1 - Z_0 \;\big|\; Z_t = z\,\big],$$

so that by the Doob–Dynkin lemma there is a (deterministic) function $v$ satisfying

$$v\big(Z_t, t\big) = \mathbb{E}\big[\,Z_1 - Z_0 \;\big|\; Z_t\,\big] = Z_1 - Z_0 \quad \text{almost surely.}$$

Hence if we re-solve the same linear ODE

$$\frac{dX_t}{dt} = v\big(X_t, t\big) \quad \text{with} \quad X_0 = Z_0,$$

the unique solution is exactly

$$X_t = Z_0 + t\,(Z_1 - Z_0) = Z_t.$$

Because the noise-free Rectified Flow minimizes the mean-squared drift error,

$$\mathbb{E}\Big\|v(Z_t, t) - \big(Z_1 - Z_0\big)\Big\|^2 = 0,$$

no further change occurs. Thus the pairing remains $(Z_0, Z_1)$. $\qquad\square$

**Proposition 2.** *Let $\pi_0$ and $\pi_1$ be two probability densities on $\mathbb{R}^D$, and let $T : \mathbb{R}^D \to \mathbb{R}^D$ be a deterministic transport map pushing $\pi_0$ to $\pi_1$ (so if $Z_0 \sim \pi_0$, then $Z_1 = T(Z_0) \sim \pi_1$). Suppose we draw a finite dataset of $N$ i.i.d. samples $\{Z_0^{(i)}, Z_1^{(i)} = T(Z_0^{(i)})\}_{i=1}^N$, and for each $i$ we also sample $m$ time-points $\{t^{(i,j)}\}_{j=1}^m \subset [0,1]$ (e.g. uniformly). Let $Z_t^{(i,j)} = (1 - t^{(i,j)}) Z_0^{(i)} + t^{(i,j)} Z_1^{(i)}$. Define the empirical loss over this "doubly indexed" dataset by*

$$L_{\mathrm{MC}}^{\mathrm{det}}(v_\Theta) \;=\; \frac{1}{N\,m} \sum_{i=1}^N \sum_{j=1}^m \left\| \big(Z_1^{(i)} - Z_0^{(i)}\big) \;-\; v_\Theta\big(Z_t^{(i,j)}, t^{(i,j)}\big) \right\|^2.$$

*Then there exists a (deterministic) vector field $v$ attaining zero loss: $L_{\mathrm{MC}}^{\mathrm{det}}(v) = 0$.*

*Proof.* Our finite dataset is

$$\mathcal{D} = \left\{ \big(Z_0^{(i)}, Z_1^{(i)}, t^{(i,j)}, Z_t^{(i,j)}\big) : i = 1, \dots, N;\ j = 1, \dots, m \right\}.$$

We may view this as drawing from the discrete joint "empirical" measure

$$\hat{P} \;=\; \frac{1}{N\,m} \sum_{i=1}^N \sum_{j=1}^m \delta_{\big(Z_0^{(i)},\, Z_1^{(i)},\, t^{(i,j)},\, Z_t^{(i,j)}\big)}.$$

For each datapoint $\big(Z_t^{(i,j)}, t^{(i,j)}\big)$ we know

$$Z_1^{(i)} - Z_0^{(i)} \;=\; \mathbb{E}\big[ Z_1 - Z_0 \;\big|\; Z_0 = Z_0^{(i)},\, Z_1 = Z_1^{(i)},\, t = t^{(i,j)},\, Z_t = Z_t^{(i,j)} \big].$$

Let $\mathcal{Z} = \bigcup_{i \in N, t \in [0,1]} (Z_t^{(i)}, t)$. Let $\mathcal{A} = \bigcap_{i \in N, t \in [0,1]} (Z_t^{(i)}, t)$.

The probability of sampling $t^{(i,j)} = a \in \mathcal{A}$ is 0, because $t$ lives in a two-dimensional distribution and the points of intersection are a countable union (maximum $N!$ intersection) of one-dimensional spaces.

$$\mathbb{E}\big[ Z_1 - Z_0 \;\big|\; Z_0 = Z_0^{(i)},\, Z_1 = Z_1^{(i)},\, t = t^{(i,j)},\, Z_t = Z_t^{(i,j)} \big] = \mathbb{E}\big[ Z_1 - Z_0 \;\big|\; t = t^{(i,j)},\, Z_t = Z_t^{(i,j)} \big].$$

Hence we can define the exact conditional-expectation vector field

$$v(z, t) \;:=\; \mathbb{E}_{\hat{P}}\big[ Z_1 - Z_0 \;\big|\; Z_t = z,\, t \big],$$

which on each of our training points $(z, t) = (Z_t^{(i,j)}, t^{(i,j)})$ satisfies

$$v\big(Z_t^{(i,j)}, t^{(i,j)}\big) = Z_1^{(i)} - Z_0^{(i)}.$$

Substituting this $v$ into the Monte Carlo loss gives, term by term,

$$\big\| Z_1^{(i)} - Z_0^{(i)} \;-\; v\big(Z_t^{(i,j)}, t^{(i,j)}\big) \big\|^2 = \big\| Z_1^{(i)} - Z_0^{(i)} \;-\; (Z_1^{(i)} - Z_0^{(i)}) \big\|^2 = 0.$$

Averaging over all $i, j$ yields $L_{\mathrm{MC}}^{\mathrm{det}}(v) = 0$.

Thus the loss can be driven exactly to zero on the finite sample by choosing $v$ to interpolate the known displacements $Z_1^{(i)} - Z_0^{(i)}$ at the sampled intermediate points $(Z_t^{(i,j)}, t^{(i,j)})$. $\qquad\square$

**Proposition 3** (Interpolating lines don't meet in high dimensions)**.** *Let $x_0, x_1 \sim \pi_0(x)$ and $y_0, y_1 \sim \pi_1(y)$, where $\pi_0$ and $\pi_1$ are probability distributions on $\mathbb{R}^d$ admitting a density. Define the linear interpolants $l_i(t_i) = (1 - t_i)x_i + t_i y_i$ for $i \in \{0, 1\}$.*

    A. *For $d \geq 2$, the lines $l_0$ and $l_1$ cross at $t = t_i = t_j \in (0, 1)$ with probability 0.*

    B. *For $d > 2$ the two lines intersect for $t_i, t_j \in (0, 1)$ with probability 0.*

*Proof.* Throughout this proof, we use the following theorem: Let $X$ be a random variable with a continuous probability distribution in $\mathbb{R}^n$, and let $A$ be a lower-dimensional subset of $\mathbb{R}^n$. Since $\mathbb{P}$ admits a density, it follows that it is absolutely continuous wrt to the Lebesgue measure $\lambda$ then as $\lambda(A) = 0$ by absolute continuity, we have that $\mathbb{P}(X \in A) = 0$.

**A.** Suppose the lines $l_0(t)$ and $l_1(t)$ intersect at some $t = \hat{t} \in (0, 1)$. This implies

$$l_0(\hat{t}) = l_1(\hat{t}),$$

which expands to

$$(1 - \hat{t})x_0 + \hat{t}y_0 = (1 - \hat{t})x_1 + \hat{t}y_1.$$

Rearranging terms, we find

$$y_1 = \frac{1 - \hat{t}}{\hat{t}}(x_0 - x_1) + y_0.$$

To compute the probability of such an intersection, note that $y_1$ must lie exactly on the affine subspace defined by the above equation, which is a one-dimensional line segment in $\mathbb{R}^d$.

The joint probability of the points $(x_0, x_1, y_0, y_1)$ can be written as

$$\mathbb{P}(l_0(t) \text{ intersects } l_1(t) \text{ for } t \in (0, 1)) = \mathbb{P}(x_0, x_1, y_0) \cdot \mathbb{P}(y_1 = \frac{1 - \hat{t}}{\hat{t}}(x_0 - x_1) + y_0 \mid x_0, x_1, y_0).$$

Since $\pi_1(y)$ is continuous and differentiable, the probability density of $y_1$ lying on any lower-dimensional subspace (e.g., a line segment) in $\mathbb{R}^d$, with $d > 2$, is zero. Therefore,

$$\mathbb{P}(y_1 = \frac{1 - \hat{t}}{\hat{t}}(x_0 - x_1) + y_0 \mid x_0, x_1, y_0) = 0,$$

which implies

$$\mathbb{P}(l_0(t) \text{ intersects } l_1(t) \text{ at } t = \hat{t} \text{ for } t \in (0, 1)) = 0.$$

Hence, the lines $l_0(t)$ and $l_1(t)$ intersect with probability zero for $t \in (0, 1)$. **B.** Suppose the lines $l_0(t_0)$ and $l_1(t_1)$ intersect at some $t_0 = \hat{t}_0, t_1 = \hat{t}_1$ for $t_0, t_1 \in (0, 1)$. This implies

$$l_0(\hat{t}_0) = l_1(\hat{t}_1),$$

which expands to

$$(1 - \hat{t}_0)x_0 + \hat{t}_0 y_0 = (1 - \hat{t}_1)x_1 + \hat{t}_1 y_1.$$

Rearranging terms, we find

$$y_1 = \frac{(1 - \hat{t}_0)x_0 - (1 - \hat{t}_1)x_1 + \hat{t}_0 y_0}{\hat{t}_1}.$$

which for $\hat{t}_0, \hat{t}_1 \in (0, 1)$ we have that $y_1$ would belong to a 2D surface. Just like before we have:

$$\mathbb{P}(l_0(t_0) \text{ intersects } l_1(t_1) \text{ for } \hat{t}_i \in (0, 1)) = \mathbb{P}(x_0, x_1, y_0) \cdot$$

$$\mathbb{P}\left(y_1 = \frac{(1 - \hat{t}_0)x_0 - (1 - \hat{t}_1)x_1 + \hat{t}_0 y_0}{\hat{t}_1} \,\middle|\, x_0, x_1, y_0\right) = 0.$$

$\square$

## D Counter Example for straightness after one iteration

**Proposition 4** (Limitations of ReFlow Iterations on Noiseless Interpolants)**.** *Let $\pi_0, \pi_1 \subset \mathbb{R}^D$, with $D > 2$. Let $(Z_0, Z_1) = $ `1-ReFlow`$(X_0, X_1)$ denote the coupling obtained after one ReFlow iteration. **Under the assumption that the interpolant $p(z, t) = (1 - t)z + tT(z)$ is injective in $z$ for each** $t$, there exists a vector field $\hat{v}_1(x_t, t)$ such that performing a second ReFlow iteration using $(Z_0, Z_1)$ yields the same coupling:*

$$\text{2-ReFlow}(X_0, X_1) = (Z_1, Z_0).$$

*Moreover, this second flow $\hat{v}_1(x_t, t)$ generates straight-line paths and achieves zero loss. Therefore, further ReFlow iterations do not alter the couplings.*

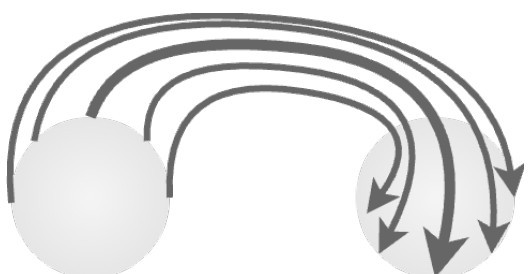

Figure 8: 180° rotation being realized by a continuous vector field.

*Annotated Proof.* Let $v_0(x_t, t)$ be the learned vector field after CFM, with Assumption 1. From Arnold [1992], the transport map $T(z_0) = z_0 + \int_0^1 v(z_t, t)dt$ with $z_0 \in \pi_0$ is injective.

**Step 1: Injectivity of $T$**

- *Assumption:* $T$ is injective (guaranteed by properties of $v$, e.g., Lipschitz, linear growth).

- *Potential Failure:* If $T$ is not injective, the argument fails immediately.

**Step 2: Injectivity of the Interpolant $p(z, t)$**

We claim that $p(z, t) = (1 - t)z + tT(z)$ is also injective in $z$ for each $t \in [0, 1]$.

- *Proof (by contradiction):* Suppose $p(z_0^{(1)}, \hat{t}) = p(z_0^{(2)}, \hat{t})$ for some $\hat{t}$ and $z_0^{(1)} \neq z_0^{(2)}$.

- Rearranging:

$$T(z_0^{(2)}) - T(z_0^{(1)}) = \frac{1 - \hat{t}}{\hat{t}}(z_0^{(1)} - z_0^{(2)})$$

- This statement doesn't really contradict with $T$ injectivity.

- *Failure Point:* As shown in Counterexample 1, this is **not always true**. For example, if $T$ is a rotation plus translation, $p(z, t)$ can fail to be injective even if $T$ is injective.

One might argue that such a transport map is unlikely to be learned in practice; however, this is not the point. Our argument is purely theoretical: injectivity of $T$ does not imply injectivity of $I(x_0, T(x_0), t)$, and thus 1-ReFlow is insufficient even under standard regularity assumptions.

**Step 3: Construction of Inverse and New Vector Field**

Assuming injectivity, we can define an inverse $f^{-1}(z_t, t) = z_0$, and then set $v_1(z_t, t) = T(f^{-1}(z_t, t)) - f^{-1}(z_t, t)$.

- *Dependency:* This construction **only works if $f^{-1}$ exists**, i.e., if $p(z, t)$ is injective.

- *Failure Point:* If $p(z, t)$ is not injective, $f^{-1}$ is not well-defined, and this construction fails.

**Step 4: Straight-Line Paths and Zero Loss**

Because $z_1 - z_0$ is uniquely determined by $z_t$, we have $\int_0^1 \mathbb{E}[\text{Var}(Z_1 - Z_0 | Z_t)] = 0$, implying straight paths and zero loss.

- *Dependency:* Again, relies on the injectivity of $p(z, t)$.

$\square$

**Counter Example 1.** *Let $\pi_0 \sim \mathcal{N}(0, \boldsymbol{I}_d)$ and $\pi_1 \sim \mathcal{N}(5, \boldsymbol{I}_d)$, and let $T(x_0) = R_{180°} x_0 + 5$, where $R_{180°}$ is a $180°$ rotation. $T$ is injective, but the interpolant $I(z_0, T(z_0), t) = (1 - t)z_0 + tT(z_0)$ is **not** injective (distinct $z_0$ can map to the same $x_t$ for some $t$). Thus, $f(x_t, t) = x_0$ is not well-defined, and the construction of $v_{new}(x_t, t)$ fails.*

| Step | Assumption Needed | Where It Can Fail |
|------|-------------------|-------------------|
| 1 | $T$ injective | Pathological $T$ |
| 2 | $p(z, t)$ injective for $t$ | Nonlinear $T$ (e.g., rotations) |
| 3 | $f^{-1}$ exists | $p(z, t)$ not injective |
| 4 | Unique $z_0$ for each $z_t$ | $p(z, t)$ not injective |

Table 4: Summary of dependencies and failure points in the proof.

**Final Note.** **This proposition is only valid under the additional assumption that the interpolant $p(z, t)$ is injective in $z$ for each $t$.** The counterexample demonstrates that this is not always the case, even when $T$ is injective. Therefore, care must be taken before applying this argument in general settings. □

# E First iteration of SBM Asymmetry

To test whether our gradient-variance diagnostic extends beyond Gaussians, we trained four U-Net models to learn the transport map from $\pi_0 = \mathcal{N}(0, \boldsymbol{I}_d)$ to $\pi_1 \sim$ CIFAR-10, using both noiseless and noisy interpolants (see for experimental details Appendix G). In all settings (Figure 9), the backward field's gradient variance peaks at $t = 0$, where CIFAR-10 appears as input. Conversely, the forward field's variance peaks at $t = 1$, when CIFAR-10 appears as the output.

To assess generation stability, we repeatedly integrated 64 fixed datapoints across 100 training checkpoints, sampling at regular intervals near the FID/NLL optima (Table 5). We report the resulting sample variances in the table's fourth column:

$$\text{Var}[x_1^{\text{gen}}] \quad \text{and} \quad \text{Var}[x_0^{\text{gen}}],$$

which show consistently higher variability in the forward direction, suggesting reduced robustness during forward-time sampling.

**Observation 1.** *The observed difference in sample variance across vector fields optimized near the optima may potentially impact subsequent iterations of SBM [Shi et al., 2024]. However, precisely characterizing how this affects theoretical bounds is non-trivial and is left as a promising avenue for future research.*

Table 5: Comparison of FID, NLL, and sample variance for CFM($\sigma = 0.05$) (first iteration of SBM) and CFM (Forward and Backward). The backward vector field exhibits approximately 10 times lower variance in integrated samples compared to the forward vector field, despite both endpoints being standardized. This highlights an inherent asymmetry in sampling stability between forward and backward flows.

|  | Direction | FID↓ | Scaled -NLL↓ | Variance ± Std. Error |
|--|-----------|------|--------------|------------------------|
| **CFM($\sigma = 0.05$)** | Backward | – | 1.423564 | $0.01647 \pm 0.00024$ |
|  | Forward | 4.250 | – | $0.36179 \pm 1.23 \times 10^{-5}$ |
| **CFM** | Backward | – | 1.426562 | $0.01414 \pm 0.00018$ |
|  | Forward | 4.199 | – | $0.36177 \pm 1.23 \times 10^{-5}$ |

# F Synthetic Experiments Details

**Experiment Setting** We evaluate generative models on synthetic datasets in dimensions 3 and 50. Each dataset is constructed by sampling from a Gaussian Mixture Model (GMM) with randomly initialized means and covariances, following our implementation in `generate_datasets.py`. The

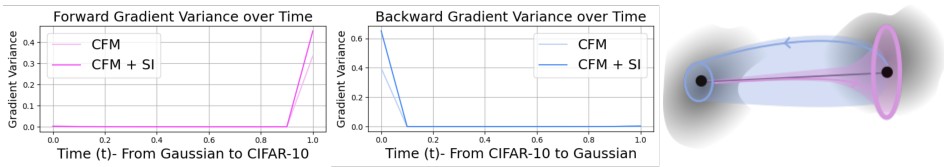

Figure 9: Comparison of gradient variance for the forward and backward passes in CFM and CFM($\sigma = 0.05$) (first iteration of SBM). The schematic on the right offers intuition for how gradient variance may influence sample variance during integration. Notably, the backward-pass gradient variance peaks near the CIFAR-10 endpoint, yet the resulting sample variance is lower compared to the forward pass. This is possible, for instance, if the forward field increases linearly while the backward field decreases linearly. These effects are quantitatively reflected in Table 1.

source distribution is standard normal, and the target is the GMM. We compare Conditional Flow Matching (CFM) and CFM with Stochastic Interpolation (CFM($\sigma = 0.05$)), both implemented as neural ODEs with time-conditioned MLP (Three-layer MLP, width 64, SELU activations) vector fields. Models are trained and evaluated on both in-sample (training) and out-of-sample (test) data. All metrics are computed as described below. We have used resources from Feydy et al. [2019] to compute the distances.

**Metric Descriptions**

- **Log Probability (LogProb):** Measures the average log-likelihood of generated samples under the target GMM distribution. Lower values indicate a better fit the modes.

- **Maximum Mean Discrepancy (MMD):** A kernel-based statistical distance between two distributions, here computed using a Gaussian kernel. Lower values indicate better sample quality.

- **Sinkhorn Distance:** An entropy-regularized approximation of the Wasserstein (optimal transport) distance between empirical distributions, computed using the Sinkhorn algorithm. Lower values indicate closer distributions.

**Note on Log-Likelihood Values**    To improve readability and avoid confusion, we report and plot the **positive** values of the log-likelihood (LogProb) throughout this paper, rather than the conventional negative log-likelihood (NLL). This allows for a more intuitive comparison, where lower values indicate better model performance.

It is important to note that log-likelihood (LogProb) primarily rewards models that generate samples close to the high-density regions (modes) of the target distribution, rather than accurately capturing the overall shape or support of the distribution. As a result, models that concentrate samples around the modes can achieve high log-likelihood scores even if they do not match the full distribution well. This should be kept in mind when interpreting LogProb values alongside other distance-based metrics.

Table 6: Comparison of CFM and CFM($\sigma = 0.05$) across dimensions 3 and 50.

| Dimension | | 3 | | | | 50 | | | |
|---|---|---|---|---|---|---|---|---|---|
| | | **Gen** | **Mem** | **True** | **Data** | **Gen** | **Mem** | **True** | **Data** |
| CFM | LogProb | 4.0150 ±0.0032 | 4.0156 ±0.0032 | 4.1330 ±0.031 | 4.0155 ±0.035 | 54.8299 ±0.015 | 53.6502 ±0.26 | 52.5094 ±0.0833 | 53.6244 ±0.014 |
| | MMD | 0.0034 ±0.0005 | $1.758 \times 10^{-6}$ | 0.0014 ±0.0001 | 0.0032 ±0.0002 | 0.0021 ±0.0001 | $9.089 \times 10^{-6}$ | 0.0020 ±2e-09 | 0.0019 ±0.0001 |
| | Sinkhorn | 0.0730 ±0.0054 | $1.411 \times 10^{-5}$ | 0.0637 ±0.002 | 0.0790 ±0.004 | 15.1900 ±0.162 | 0.0045 | 14.3221 ±0.110 | 15.7400 ±0.130 |
| CFM($\sigma = 0.05$) | LogProb | 4.1270 ±0.0010 | 4.0960 ±0.016 | 4.1330 ±0.0008 | 4.0155 ±0.0013 | 54.7220 ±0.14 | 53.8890 ±0.16 | 52.5094 ±0.11 | 53.6244 ±0.26 |
| | MMD | 0.0018 ±0.00048 | $3.105 \times 10^{-5}$ | 0.0014 ±0.0003 | 0.0032 ±0.0002 | 0.0020 ±0.0001 | $6.09 \times 10^{-5}$ | 0.0020 ±3e-09 | 0.0019 ±0.0001 |
| | Sinkhorn | 0.0680 ±0.0015 | $3.557 \times 10^{-4}$ | 0.0637 ±0.007 | 0.0790 ±0.004 | 15.1689 ±0.170 | 0.0304 | 14.3221 ±0.220 | 15.7400 ±0.230 |

**How to read the table?**   This table summarizes several metrics for model evaluation. For readers unfamiliar with these results, here is how to interpret them:

Consider the case for dimension 3. For the **Gen** (generated) column, we want the log-likelihood value to be closer to the **True** value rather than the **Data** value. If the generated value is closer to the data, it indicates memorization, rather than true generalization. For example, in CFM, the generated value is much closer to the data than the true value, suggesting memorization. This discrepancy arises because negative log-likelihood (NLL) tends to favor models that sample near the training data modes, rather than those that capture the full distribution.

Similarly, in the **Mem** (memorization) column, a value close to the data again indicates overfitting to the training points.

For the **MMD** and **Sinkhorn** metrics, these measure distances between newly generated trajectories (starting from training points) and the pairings from previous iterations. In both MMD and Sinkhorn, we observe that CFM($\sigma = 0.05$) memorizes roughly ten times less than standard CFM, indicating better generalization.

### F.1 Asymmetry in Gaussian mixtures

**Experiment Settings**   We systematically investigated the Conditional Flow Matching (CFM) and CFM($\sigma = 0.05$) approaches for learning mappings between mixtures of Gaussian distributions (GMM-to-GMM). In each experiment, both the source and target distributions were Gaussian mixtures, with the number of components for each varied across the grid: $\{1, 2, 4, 8, 16, 32, 64\}$. The neural network architecture for the vector field was a multilayer perceptron (MLP) with configurable width ($w$, e.g., 64 by default) and input dimension ($d$, e.g., 10). Each cell in the results grid corresponds to a specific pair of source and target GMM component counts, allowing us to analyze the effect of distribution complexity on learning dynamics and gradient variance. Training was performed for up to 50,000 epochs using a batch size of 128 and a learning rate of $10^{-3}$, with integration performed via Neural ODEs.

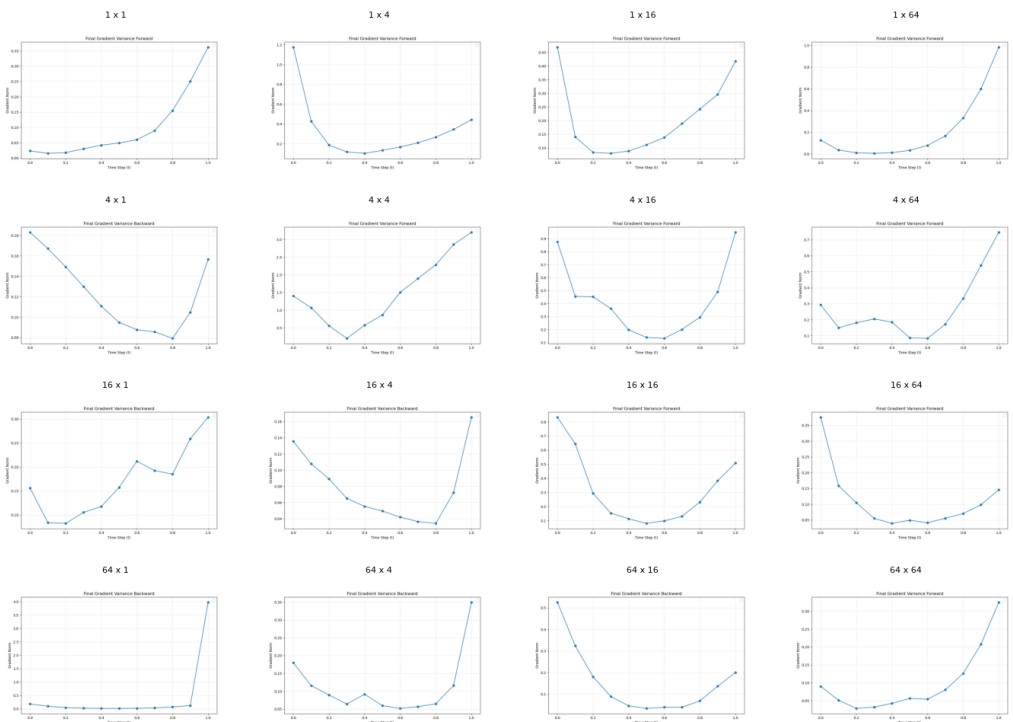

Figure 10: CFM in 10 dimensions. Each title indicates the number of Gaussian components in the source distribution multiplied by the number in the target distribution (e.g., 4×16 means 4 modes at source, 16 at target).

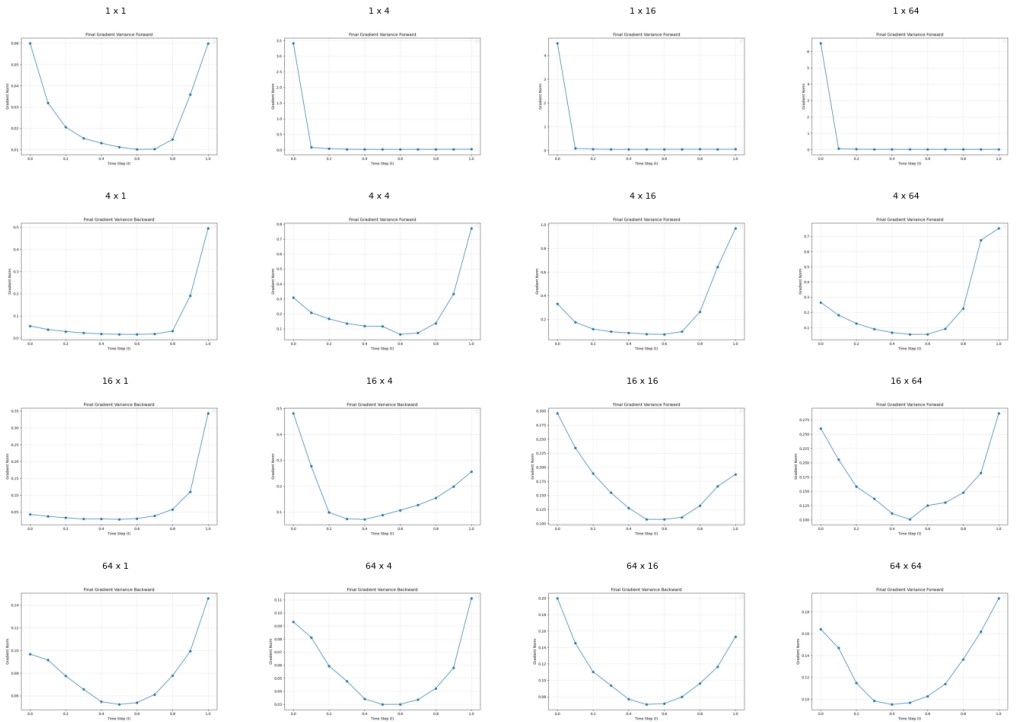

Figure 11: CFM combined with stochastic interpolants for mixture-to-mixture transport in 10 dimensions. Each panel is labeled as source modes × target modes.

We can notice that the variance of the gradients looks more stable in the case of the Figure 11 (stochastic interpolants), than in the case of Figure 10 (noisless).

## G  CIFAR-10 Experiment Details

**Model architecture.**    All experiments used a U-Net-based neural network (`UNetModelWrapper`) with the following configuration: input shape $(3, 32, 32)$, base channels $128$, 2 residual blocks per level, channel multipliers $[1, 2, 2, 2]$, attention at $16 \times 16$ resolution (4 heads, 64 head channels), and dropout rate $0.1$. The model is wrapped in a Neural ODE solver (Euler method).

**Training.**    Models were trained on the CIFAR-10 training set, using random horizontal flips and normalization to $[-1, 1]$. Optimization used Adam with learning rate $2 \times 10^{-4}$, batch size $128$, gradient clipping at $1.0$, and a linear warmup over the first $5,000$ steps. Each run used $400,001$ steps (unless otherwise noted), with exponential moving average (EMA) of model weights ($0.9999$ decay). Checkpoints were saved every $20,000$ steps. All experiments used $4$ data loader workers and CUDA if available.

**Flow objectives.**    We used Conditional Flow Matching (CFM, `-model cfm`), Schrödinger Bridge Matching (SBM, `-model sbm`), and other variants.

**Bidirectional setup.**    Both forward (Gaussian $\rightarrow$ CIFAR-10) and backward (CIFAR-10 $\rightarrow$ Gaussian) models were trained independently with identical hyperparameters. For the forward model, the source is standard Gaussian noise and the target is real images; for the backward model, the roles are swapped.

**Hardware and runtime.**    All CIFAR-10 experiments were conducted on a compute cluster equipped with NVIDIA A10 GPUs (24 GB VRAM, CUDA 12.2). Each training run was allocated a single A10 GPU and typically ran for 24 hours to reach 240,000 optimization steps. These resources enabled efficient training of both forward and backward models at the scale reported in the main text.

### G.1 FID and Interpolant Details

For Figure 9, we trained four vector fields (two forward and two backward), each using a different interpolant:

- **CFM:** The interpolant is deterministic,

$$x_t = (1 - t)x_0 + tx_1.$$

- **CFM**$(\sigma = 0.05)$**:** The interpolant includes stochastic noise,

$$x_t = (1 - t)x_0 + tx_1 + \sigma\sqrt{t(1-t)}Z,$$

where $Z \sim \mathcal{N}(0, I)$ and $\sigma = 0.01$.

The models were evaluated as follows:

- CFM forward FID: **4.199**
- CFM backward scaled NLL: **1.426**
- CFM$(\sigma = 0.05)$ forward FID: **4.250**
- CFM$(\sigma = 0.05)$ backward scaled NLL: **1.423**

### G.2 Does adding powers of $t$ help models learn better?

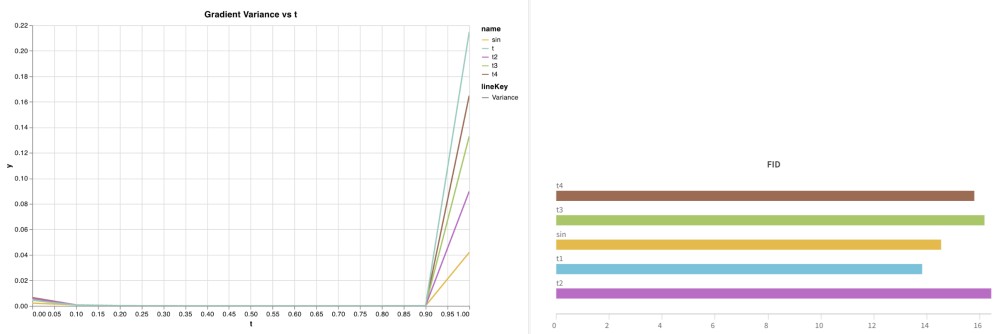

Figure 12: Variance not necessarily correlated with performance. I was wondering if it is worth saying that sinusoidal time embedding is better at representing function s like $q(x) = \frac{1}{1+x}$, than polynomials. (Taylor vs Fourier)

## H  CelebA Experiment Details

In Table 7, we computed FID on both the training and validation sets and observed similar values, which could, in principle, indicate the trade-off between memorization and generalization. However, the CelebaOt dataset Korotin et al. [2021] is itself generated by another model, and with only 50k generated samples it likely contains overlapping images. Consequently, the training and validation splits may share duplicates, making it difficult to draw strong conclusions from the FID scores. We include these results merely as evidence that all models were trained to a satisfactory degree.

| Setting | Method | FID ↓ |
|---------|--------|-------|
| Shuffled | CFM($\sigma = 0$) | 34.36 |
|          | CFM($\sigma = 0.05$) | 32.31 |
| ReFlow | CFM($\sigma = 0$) | 31.42 |
|        | CFM($\sigma = 0.05$) | 42.38 |

Table 7: FID scores for the 50k dataset-side setting. Lower is better.

