# OpenReview forum: "Gradient Variance Reveals Failure Modes in Flow-Based Generative Models"
_NeurIPS.cc/2025/Conference — NeurIPS 2025 spotlight_

### Official Review · Reviewer_EgrR · 2025-06-28

**Clarity:** 2
**Significance:** 2
**Originality:** 3
**Rating:** 3
**Confidence:** 3

**Summary:**

This paper investigates the causes of failure modes behind Rectified Flows and Schrödinger Bridge. This paper introduces a gradient-variance diagnostic to measure the transport errors of the learned vector fields. Utilizing the Gaussian transport, the paper shows that the neural networks fail to obtain strict convergence. Moreover, the paper proves that Reflow with noiseless interpolants stagnate when arriving at straight couplings, and memorize when trained on deterministic couplings. The paper also provides a counterexample to show that 1-Reflow could fail. To address to these drawbacks, the paper claims that noise can alleviate these two effects. The paper experiments on synthetic dataset and CIFAR-10 to support this claim.

**Questions:**

1. The paper claims in the Gaussian setting that MLP cannot approximate without error. Is this really a critical issue? In experiments, other types of errors like discretization and sample errors also affects the final results, the approximation error could not be the main influencing factor.
2. What is the main point of Section 3.2? If only an observation of the sample variance of the SBM is provided, it seems that this part is not strong enough to support the main findings and could be removed.
3. The second last row of the caption of Figure 3 states that ”confidence intervals narrow with higher $\sigma$”. This seems to be not apparent in Figure 3 since the shaded regions seems to be the same width across different $\sigma$s?

**Ethical Concerns:**

["NO or VERY MINOR ethics concerns only"]

**Final Justification:**

I prefer to maintain my evaluation score. Although the authors conducted additional experiments on CelebA in the rebuttal, more discussions on the choice of $\sigma$ could be done to increase the practicality of the proposed method. Furthermore, the current manuscript needs revision on the focus of the central contribution and the clarity. Hence, in my view, the current manuscript is not yet ready for publication.

**Limitations:**

The paper addressed most of the limitations. As a supplement, it remains to be examined how much drawback reflected by the Gaussian transport will appear in general flow matching methods, which is widely used in various fields.

**Paper Formatting Concerns:**

No.

**Quality:**

2

**Strengths And Weaknesses:**

Strengths:
1. The perspective to study gradient variance to provide understandings on the limitations of neural networks and Rectified Flows is novel.
2. The paper provides the counter example to show that 1-Reflow may fail in specific cases, which contributes to the study of Rectified Flows.

Weaknesses:
1. The clarity of the paper writing can be improved. The title focuses on the “gradient variance”. However, in section 4, most of the claims do not closely relate to the “gradient variance”, making the focus of the paper being obscured. In line 177-183, the key takeaways include multiple serial code (A)s and (B)s. In Proposition 1, the text and the formulas in line 157-159 are mixed together with multiple “and”s.
2. It remains to be examined how much drawback reflected by the synthetic Gaussian transport counter examples will truly appear in general flow matching methods, which is widely used in various fields with acceptable results.
3. The paper only presents potential understandings on the stagnation and memorization issues, but it does not provide new methods or ideas to tackle the issues while preserving the benefits of noiseless methods like Rectified Flows. Preferring stochastic flow matching is not a permanent solution, since deterministic and straight paths enjoy other advantages like faster simulation and smaller discretization errors. Moreover, the novelty of utilizing noise in generation is not that significant.
4. It would be beneficial to experiment on more datasets to show the of the broad applicability of the proposed gradient variance study.

---

> ### Author Rebuttal · Authors · 2025-07-29
>
> We thank the reviewer for their thoughtful feedback. It helped us clarify the connection between Section 4 and earlier theory, refine the gradient variance discussion in Section 3.2, and reframe the Gaussian example. We also added new CelebA experiments that support our theoretical findings, especially Proposition 2.
>
> ## Addressing weaknesses
>
> > “The clarity of the paper writing can be improved.”
>
> We thank the reviewer for this valuable feedback. We will revise the manuscript to improve clarity and better emphasize the central role of gradient variance throughout. Specifically, we will improve the exposition of Proposition 1 and revise lines 177–183 to avoid overloading the narrative with serial (A)/(B) references.
>
> We will strengthen the paper’s narrative thread and relation to gradient variance by clarifying the following schema, both textually and through an improved Figure 1:
>
> (1) Lemma 1 (Gaussian closed-form) → Prop. 1 (grad. variance bounds)
> (2) Gaussian experiments
> (1)+(2) ⇒ Conclusions (low grad.variance  ≠ OT; reflow skips intersection) → (generalizes) → Prop. 2 (finite data).
>
> > “...but no new method.”
> > “Straight paths give faster simulation and lower discretization error.”
>
> We agree that straight-line deterministic interpolants are attractive due to their simulation efficiency. While we do not propose a new method, our contribution lies in showing that this efficiency can obscure critical failure modes: in noiseless settings, models can minimize both loss and gradient variance while still learning non-optimal ($W_2$) couplings.
>
> This challenges key assumptions in Rectified Flows:
> 1. Gradient variance reflects path intersections [3,4]
> 2. ReFlow corrects such intersections via iteration [5]
> 3. Two iterations suffice for convergence [1,6]
>
> In finite-data regimes, these assumptions break down:
> - No convergence to straight paths after two steps (Counterexample 1)
> - Skipping intersections and vanishing variance where paths cross (Fig. 6)
> - Loss and variance minimized via memorization of poor couplings (Prop. 2 + CelebA experiment)
>
> While straight paths are efficient, our results show they may never be reached in practice (Fig. 6), highlighting the need for more robust transport strategies.
>
>
> > “The novelty of utilizing noise is not that significant.”
>
> We agree, and we do not claim that noise addition is novel. Instead, it plays a key role in our analysis by disrupting the conditions under which memorization arises (see Prop. 2, and CelebA experiments). The contrast between deterministic and noisy settings helps reveal structural limitations in the loss landscape of ReFlow.
>
> > “beneficial to experiment on more datasets”,  “to be examined how much drawback reflected by the synthetic Gaussian transport counter examples will truly appear in general flow matching methods”
>
> To address concerns about generalization beyond synthetic settings, we provide new experiments on the CelebA benchmark. These results directly validate Proposition 2 on high-resolution, real-world data and confirm that the observed memorization dynamics persist in complex visual domains, not just toy examples.
>
> ## Questions
>
> >  MLP cannot approximate without error. Is this really a critical issue?
>
> The main purpose of Lemma 1 is to give the closed-form OT vector field and its rotated version, which Proposition 1 builds upon. The extra fact about exact representability is not needed for the proof, but shows that even Gaussians yield functions (e.g., 1/x) hard to approximate with standard networks.
>
> > Section 3.2 role?
>
> We agree that Section 3.2 is less central to the main narrative. We will move it to the appendix and revise the surrounding text to maintain focus on the paper’s core contribution: understanding failure modes in vector field learning through gradient variance.
>
> >  Figure 3 caption
>
> Thank you for catching the error in Figure 3’s caption. We meant to convey that as $\sigma$ increases, the optimal solution shows lower variance than the vector field aligned with the pairing.
>
> ## New Experiments on CelebA
>
> To empirically validate Proposition 2, we evaluate Conditional Flow Matching (CFM) on CelebA using ground-truth OT pairings from [2]. Our goal is to test the central claim: when trained with deterministic interpolants, CFM is optimized to memorize its input pairings.
>
> To test this, we compare:
>
> - **CFM (σ = 0):** trained with deterministic interpolants
>   $x_t = (1 - t) x_0 + t x_1$
>
> - **CFM (σ = 0.05):** trained with slightly noisy interpolants
>   $x_t = (1 - t) x_0 + t x_1 + \sqrt{t(1 - t)} \, \sigma \epsilon$
>
> The small noise ($\sigma = 0.05$) is not for performance, but to break injectivity between $(x_t, t)$ and $(x_0, x_1)$—disabling memorization (Prop. 2).
>
>
> ### Adversarial Pairings: CFM Optimizes for Memorization with Deterministic Interpolants
>
> **Setup:** To directly test whether CFM with deterministic interpolants ($\sigma = 0$) is optimized to memorize suboptimal pairings, we construct an adversarial dataset:
>
> - We start from ground-truth OT pairs $(x_0, x_1)$ derived from the benchmark of Korotin et al. [2].
> - We shuffle the targets $(x_1)$, breaking the optimal structure and producing shuffled deterministic pairings $(x_0, S(x_1))$.
>
> We compare CFM (σ = 0) and CFM (σ = 0.05) on $(x_0, S(x_1))$.
>
> | Dataset     | Analysis       | Metric         | CFM (σ = 0.05) | CFM (σ = 0.0) | Advantage                         |
> | ----------- | -------------- | -------------- | -------------- | ------------- | --------------------------------- |
> | 5K Samples  | Generalization | L2 to True OT  | 34.25 ± 7.54   | 50.40 ± 16.73 | CFM (σ = 0.05) generalizes better |
> | 5K Samples  | Memorization   | L2 to Shuffled | 55.02 ± 16.51  | 28.57 ± 5.49  | CFM (σ = 0.0) memorizes more      |
> | 50K Samples | Generalization | L2 to True OT  | 30.05 ± 6.77   | 46.78 ± 14.87 | CFM (σ = 0.05) generalizes better |
> | 50K Samples | Memorization   | L2 to Shuffled | 56.48 ± 18.55  | 45.98 ± 11.85 | CFM (σ = 0.0) memorizes more      |
>
>
> **Evaluation Protocol (Lower is better):**
> We report L2 reconstruction error to  the true OT targets to assess generalization, and  the shuffled training targets to assess memorization.
>
> Since the pairings are deliberately incorrect, a low L2 error to shuffled targets indicates overfitting to suboptimal data. High L2 to these targets signals resistance to memorization.
>
> **Interpretation:**
> These results confirm Proposition 2: CFM with deterministic interpolants ($\sigma = 0$) minimizes loss by memorizing training targets—even when the pairings are arbitrarily shuffled. In contrast, CFM with noise ($\sigma = 0.05$) avoids this and generalizes better to the true OT map.
>
> **Extra comment:**
> Memorization decreases with larger datasets because the model size is fixed, making it harder to memorize all pairings. This does not contradict Proposition 2: the objective still favors memorization when possible—larger datasets limit its feasibility, not its tendency.
>
> ---
>
> ### Simulated 1-ReFlow: CFM Memorizes Its Own Outputs
>
> **Setup:** To evaluate whether iterative flow models suffer from self-reinforcing memorization, we simulate a 1-step ReFlow scenario—a core case of Proposition 2.
>
> - **Initial training:** We train a base CFM model ($\sigma = 0$) on random, non-OT pairings.
> - **Integration:** Using this model, we generate new endpoints $(x_0, x^1)$ via ODE integration.
>
> We compare CFM (σ = 0) and CFM (σ = 0.05) on $(x_0, x^1)$.
>
>
> | Dataset     | Analysis       | Metric        | CFM (σ = 0.05) | CFM (σ = 0.0) | Advantage                         |
> | ----------- | -------------- | ------------- | -------------- | ------------- | --------------------------------- |
> | 5K Samples  | Generalization | L2 to True OT | 31.35 ± 7.38   | 43.35 ± 14.21 | CFM (σ = 0.05) generalizes better |
> | 5K Samples  | Memorization   | L2 to $x^1$     | 25.08 ± 8.59   | 8.63 ± 1.76   | CFM (σ = 0.0) memorizes more      |
> | 50K Samples | Generalization | L2 to True OT | 32.54 ± 8.86   | 38.16 ± 11.37 | CFM (σ = 0.05) generalizes better |
> | 50K Samples | Memorization   | L2 to $x^1$     | 16.29 ± 4.82   | 12.76 ± 4.01  | CFM (σ = 0.0) memorizes more      |
>
>
> **Evaluation Protocol (Lower is better):**
> We report L2 reconstruction error to the true OT targets to assess generalization, and the generated training targets $(x_0, x^1)$ to assess memorization. Since the pairings come from a previous CFM model trained on random (non-OT) data, a low L2 to $x^1$ indicates that the model is overfitting to flawed past outputs. High error rates to these targets signal that the model is resisting memorization and potentially correcting earlier mistakes.
>
> **Interpretation:**
> The results in the table provide further support for Proposition 2: CFM with deterministic interpolants ($\sigma = 0$) is highly effective at memorizing the training couplings, even when those pairings are suboptimal and self-generated. In contrast, CFM ($\sigma = 0.05$), despite only a small amount of noise, avoids this memorization trap and learns to generalize toward the true OT map, despite never being directly trained on it.
>
> ---
>
> ### References
>
> [1] Saptarshi Roy, Vansh Bansal, Purnamrita Sarkar, and Alessandro Rinaldo. *2-rectifications are enough for straight flows: A theoretical insight into Wasserstein convergence.*
>
> [2] Korotin, Alexander, et al. *Do neural optimal transport solvers work? A continuous Wasserstein-2 benchmark.* NeurIPS 34 (2021): 14593–14605.
>
> [3] Fjelde, T., Mathieu, E., and Dutordoir, V. *An Introduction to Flow Matching.* MLG Blog, January 2024.
>
> [4] Gagneux, A., Martin, S., Emonet, R., Bertrand, Q., and Massias, M. *A Visual Dive into Conditional Flow Matching.* ICLR Blogposts, April 2025.
>
> [5] Liu, Xingchao, Chengyue Gong, and Qiang Liu. *Flow straight and fast: Learning to generate and transfer data with rectified flow.*
>
> [6] Lee, Sangyun, Zinan Lin, and Giulia Fanti. *Improving the training of rectified flows.* NeurIPS 37 (2024): 63082–63109.

---

> > ### Comment · Reviewer_EgrR · 2025-08-04
> >
> > I thank the authors for their detailed response and for conducting additional experiments on CelebA. Nevertheless, in my view, the current manuscript needs revision on the focus of the central contribution and the clarity, hence it is not yet ready for publication.
> >
> > For the additional experiments on CelebA in the rebuttal, I would like to see some other metrics like Frechet inception distance (FID) score of CFM ($\sigma=0$) and CFM ($\sigma=0.05$) to show the quality of image samples. Additionally, in the rebuttal of Reviewer yZa2, the authors proposed using gradient variance to tune the noise scale during training by comparing the gradient variance trained with and without noise. Is it possible to use this method to tune the noise scale $\sigma$ to obtain more significant results than $\sigma=0.05$? More such discussions on the choice of $\sigma$ with numerical validations would be appreciated to increase the practicality of the paper.
> >
> > Due to the lack of the readability and the practicality of the current manuscript, I prefer to maintain my current evaluation score.

---

> ### Author Response · Authors · 2025-08-08
>
> Dear Reviewer,
>
> Thank you for your continued input and thoughtful feedback. We agree that reporting FIDs allows our results to be more comparable / easier to validate.
>
> We just ran a quick evaluation on 10,000 samples, and the values for all models' FIDs were within 30-40, indicating that all models have relatively good image quality. We will add a table with these results in the next version of our manuscript.
>
> We are currently in the process of designing (and running) a FID-based benchmark that is suitable for our memorisation/shuffling experiments as an alternative to the precomputed OT pairings from [2]. Unfortunately, we won't be able to finish this by the end of the review period, which is today AoE. Still, we will add these in the next version of our manuscript.
>
> We also appreciate the feedback on the clarity, and do agree that restructuring is necessary to ensure more cohesion with the results, and a stronger flow/conclusion. In our final response to reviewer vzJd (above) we addressed this more thoroughly, providing more precise conclusions and connective tissue.

---

### Official Review · Reviewer_JSzR · 2025-07-02

**Clarity:** 2
**Significance:** 3
**Originality:** 3
**Rating:** 5
**Confidence:** 4

**Summary:**

The paper investigates the common causes of model failures in class of the state of the art (SOTA) fixed point conditional generative models: Rectified Flows (ReFlows), Schrödinger Bridge Matching (SBMs). The main analysis lies in the statistical investigation of the gradient variance to decide whether the resulting vector field is optimal or not. The authors evidently demonstrate counterexamples when the low gradient variance does not coincide with the optimality of the vector field. The presented examples of SOTA models fail to learn exactly the transport between two angled Gaussians. The authors paid special attention to ReFlow model indicating its stagnation on straight couplings tending to memorize deterministic pairings. The noisy interpolants can mitigate these issues on toy examples. The empirical verification of the claimed results provided for conditional flow matching on CIFAR-10 dataset.

**Questions:**

1) How ReFlows would behave on datasets given the varying dataset size, dimension, noise?
2) Can additional regularization (eg. KL on distributions) and learning rate/batch size scheduling avoid memorization effect or suboptimal vector fields?
3) Can you provide some more (counter)examples on CFN suboptimality while achieving a stationary point on loss curve for SBM?

**Ethical Concerns:**

["NO or VERY MINOR ethics concerns only"]

**Final Justification:**

Authors add additional experiments and theoretical justification sum up to a more solid work with a more extensive set of experiments, so I’m willing to increase my final rating to 5: Accept.

**Limitations:**

1) The work needs extended experiments with varying dataset sizes, dimensions and (likely) domain types to verify the consistence of provided claims.
2) The work lack some thorough analysis especially towards SBMs and optimal transport (OT).
3) The work indicates concrete limitations but hardly discuss how to fix it.

**Paper Formatting Concerns:**

No problems

**Quality:**

3

**Strengths And Weaknesses:**

The work provides some rigorous theoretical results on counterexamples for conditional flow matching (CFN) including SBMs and ReFlows.
The experiments on Gaussian transport and CIFAR-10 support the claims in general.
The work reconsiders very common assumptions in CFNs providing counterexamples.

---

> ### Author Rebuttal · Authors · 2025-07-29
>
> Thank you for the constructive and thoughtful review. We appreciate your interest in ReFlow’s behavior under different settings. Your feedback helped us clarify the broader implications of our theoretical findings and inspired new experimental directions.
>
>
> ## Addressing Weaknesses & Limitations
> > "The work needs extended experiments with varying dataset sizes, ...”
> > “How ReFlows would behave on datasets given ...”
> > “Some more (counter)examples on CFN suboptimality while achieving a stationary point on loss”
>
> We now include new experiments on CelebA OT pairs from [1] that directly validate Proposition 2 in both adversarial and ReFlow-style settings, varying dataset size and noise scale. These settings allow us to probe the relationship between vector field quality and interpolation strategy, even at stationary points of the loss curve.
>
> > “The work lacks some thorough analysis, especially towards SBMs and optimal transport (OT).”
>
> To address this, we evaluate CFM and SBM-like interpolants across four types of pairing:
> 1. Random pairings
> 2. Ground-truth OT pairings
> 3. Shuffled OT pairings: $(x_0, \mathcal{S} (\text{OT}(x_0)))$ , where $\mathcal S$ is a shuffling function
> 4. CFM-integrated pairings
>
> We assess how closely learned couplings align with the OT map and the pairings used during training. These experiments isolate whether models genuinely recover OT or merely memorize their training signal. We look forward to your thoughts on these results.
>
> > Work indicates concrete limitations but hardly discusses how to fix it.
>
> We believe highlighting ReFlow’s limitations is a necessary first step toward addressing its failure modes. Existing literature largely assumes idealized conditions (e.g., infinite data, perfect discretization), while our work shows these assumptions break down in practice. Our added noise experiments—simple yet theoretically grounded—offer a promising direction for improving generalization and vector field quality. We hope this opens the door for further study.
>
>
> ## Questions
>
> >  Can regularization (e.g., KL, LR scheduling) mitigate suboptimality?
>
> We use dropout and gradient clipping, but not weight decay or KL-based regularization. While such techniques can help smooth solutions, they cannot eliminate the root issue: the learning objective permits zero-loss, zero-gradient-variance solutions that perfectly memorize mismatched pairings (see Proposition 2).
> Crucially, our findings arise even without strong regularization. The contrast between CFM (noiseless) and CFM (noisy) models—trained identically—highlights how interpolant choice fundamentally affects learning, independent of regularization strategy.
>
> ---
>
>
> ## New Experiments on CelebA
>
> To empirically validate Proposition 2, we evaluate Conditional Flow Matching (CFM) on CelebA using ground-truth OT pairings from Korotin et al. [1]. Our goal is to test the central claim: when trained with deterministic interpolants, CFM is optimized to memorize its input pairings.
>
> To test this, we compare:
>
> - **CFM (σ = 0):** trained with deterministic interpolants
>   $x_t = (1 - t) x_0 + t x_1$
>
> - **CFM (σ = 0.05):** trained with slightly noisy interpolants
>   $x_t = (1 - t) x_0 + t x_1 + \sqrt{t(1 - t)} \, \sigma \epsilon$
>
> The small amount of noise ($\sigma = 0.05$) is not introduced for performance reasons but to break the injectivity between $(x_t, t)$ and $(x_0, x_1)$—disabling the memorization pathway outlined in Proposition 2. Both models use identical U-Net architectures.
>
> ---
>
> ### Adversarial Pairings: CFM Optimizes for Memorization with Deterministic Interpolants
>
> **Setup:** To directly test whether CFM with deterministic interpolants ($\sigma = 0$) is optimized to memorize suboptimal pairings, we construct an adversarial dataset:
>
> - We start from ground-truth OT pairs $(x_0, x_1)$ derived from the benchmark of Korotin et al. [2].
> - We shuffle the targets $(x_1)$, breaking the optimal structure and producing random pairings $(x_0, S(x_1))$ with no coherent transport semantics.
>
> We compare CFM (σ = 0) and CFM (σ = 0.05) on $(x_0, S(x_1))$.
>
> | Dataset     | Analysis       | Metric         | CFM (σ = 0.05) | CFM (σ = 0.0) | Advantage                         |
> | ----------- | -------------- | -------------- | -------------- | ------------- | --------------------------------- |
> | 5K Samples  | Generalization | L2 to True OT  | 34.25 ± 7.54   | 50.40 ± 16.73 | CFM (σ = 0.05) generalizes better |
> | 5K Samples  | Memorization   | L2 to Shuffled | 55.02 ± 16.51  | 28.57 ± 5.49  | CFM (σ = 0.0) memorizes more      |
> | 50K Samples | Generalization | L2 to True OT  | 30.05 ± 6.77   | 46.78 ± 14.87 | CFM (σ = 0.05) generalizes better |
> | 50K Samples | Memorization   | L2 to Shuffled | 56.48 ± 18.55  | 45.98 ± 11.85 | CFM (σ = 0.0) memorizes more      |
>
>
> **Evaluation Protocol (Lower is better):**
> We report L2 reconstruction error to  the true OT targets to assess generalization, and  the shuffled training targets to assess memorization.
>
> Since the pairings are deliberately incorrect, a low L2 error to shuffled targets indicates overfitting to suboptimal data. High L2 to these targets signals resistance to memorization.
>
> **Interpretation:**
> The results directly support Proposition 2: CFM with deterministic interpolants ($\sigma = 0$) minimizes loss by tightly memorizing its training targets, even when they are arbitrarily shuffled and non-optimal. In contrast, CFM with small noise ($\sigma = 0.05$) resists this memorization trap and generalizes significantly better to the true OT targets.
>
> **Extra comments:**
> - Memorization decreases with larger datasets, as expected, since model capacity is fixed, it becomes harder to memorize all pairings in the 50K setting.
> - However, this does not contradict Proposition 2: the optimization still favors memorization when possible. Larger datasets simply make perfect memorization less feasible, not less likely as an objective.
>
> ---
>
> ### Simulated 1-ReFlow: CFM Memorizes Its Own Outputs
>
> **Setup:** To evaluate whether iterative flow models suffer from self-reinforcing memorization, we simulate a 1-step ReFlow scenario—a core case of Proposition 2.
>
> - **Initial training:** We train a base CFM model ($\sigma = 0$) on random, non-OT pairings.
> - **Integration:** Using this model, we generate new endpoints $(x_0, x^1)$ via ODE integration.
>
> We compare CFM (σ = 0) and CFM (σ = 0.05) on $(x_0, x^1)$.
>
>
> | Dataset     | Analysis       | Metric        | CFM (σ = 0.05) | CFM (σ = 0.0) | Advantage                         |
> | ----------- | -------------- | ------------- | -------------- | ------------- | --------------------------------- |
> | 5K Samples  | Generalization | L2 to True OT | 31.35 ± 7.38   | 43.35 ± 14.21 | CFM (σ = 0.05) generalizes better |
> | 5K Samples  | Memorization   | L2 to $x^1$     | 25.08 ± 8.59   | 8.63 ± 1.76   | CFM (σ = 0.0) memorizes more      |
> | 50K Samples | Generalization | L2 to True OT | 32.54 ± 8.86   | 38.16 ± 11.37 | CFM (σ = 0.05) generalizes better |
> | 50K Samples | Memorization   | L2 to $x^1$     | 16.29 ± 4.82   | 12.76 ± 4.01  | CFM (σ = 0.0) memorizes more      |
>
>
> **Evaluation Protocol (Lower is better):**
> We report L2 reconstruction error to the true OT targets to assess generalization, and the generated training targets $(x_0, x^1)$ to assess memorization. Since the pairings come from a previous CFM model trained on random (non-OT) data, a low L2 to $x^1$ indicates that the model is overfitting to flawed past outputs. High error rates to these targets signal that the model is resisting memorization and potentially correcting earlier mistakes.
>
> **Interpretation:**
> The results in the table provide further support for Proposition 2: CFM with deterministic interpolants ($\sigma = 0$) is highly effective at memorizing the training couplings, even when those pairings are suboptimal and self-generated. In contrast, CFM ($\sigma = 0.05$), despite only a small amount of noise, avoids this memorization trap and learns to generalize toward the true OT map, despite never being directly trained on it.
>
> ---
>
> ### References
>
> [1] Korotin, Alexander, et al. *Do neural optimal transport solvers work? A continuous Wasserstein-2 benchmark.* NeurIPS 34 (2021): 14593–14605.

---

> > ### Comment · Reviewer_JSzR · 2025-08-06
> >
> > Thank you for your response to the review. Your additional experiments and theoretical justification sum up to a more solid work with a more extensive set of experiments, so I’m willing to increase my final rating to 5: Accept.

---

### Official Review · Reviewer_yZa2 · 2025-07-03

**Clarity:** 3
**Significance:** 3
**Originality:** 3
**Rating:** 4
**Confidence:** 4

**Summary:**

This paper investigates the fundamental limitations of flow-based generative models, particularly Rectified Flows and Schrödinger Bridge Matching, through a novel gradient variance analysis framework. The authors introduce a diagnostic tool that measures how learned vector fields induce transport errors by analyzing the variance of gradients during optimization. Their main theoretical contributions include proving that standard neural network architectures cannot exactly represent even simple Gaussian-to-Gaussian optimal transport, demonstrating that Rectified Flows can stagnate when reaching straight couplings and memorize deterministic pairings rather than learning optimal transport, and showing that stochastic interpolants help break these failure modes. The work challenges the common assumption that straight transport paths are always optimal and provides both theoretical insights and empirical validation on synthetic Gaussian data and CIFAR-10.

**Questions:**

- Can you provide results on additional real-world datasets beyond CIFAR-10? Your theoretical claims about fundamental limitations would be much stronger with broader empirical validation. Specifically, how do your findings hold for higher-resolution images, different data modalities, or more complex transport scenarios?
- While the gradient variance diagnostic is theoretically interesting, how can practitioners actually use this tool? Can you provide concrete guidelines for when the diagnostic suggests switching to stochastic interpolants or adjusting noise levels? A practical algorithm that leverages your insights would significantly strengthen the contribution.
- Comparison with recent advances: How do your findings relate to more recent developments in flow matching, such as consistency models or recent improvements to Rectified Flows? Have any of these approaches inadvertently addressed the memorization issues you identify?

**Ethical Concerns:**

["NO or VERY MINOR ethics concerns only"]

**Final Justification:**

I maintain my original rating and leaning towards acceptance for this work.

**Limitations:**

Yes, the authors adequately address limitations in their conclusion, acknowledging the restricted experimental scope to CIFAR-10 and the preliminary nature of their SBM analysis.

**Paper Formatting Concerns:**

No.

**Quality:**

3

**Strengths And Weaknesses:**

# Strengths
- I find the theoretical framework quite compelling. The gradient variance diagnostic is genuinely novel and provides insights that pure loss analysis misses - the counterintuitive finding that low variance doesn't guarantee optimal transport is particularly valuable. The mathematical rigor is impressive, especially Lemma 1's proof that MLPs cannot exactly represent the matrix inverse terms in optimal transport vector fields, and Proposition 2's formal characterization of memorization in finite datasets.
- I think the paper makes important corrections to recent literature. Counte-rexample 1 effectively refutes claims about 1-Reflow sufficiency using rotation-based transport maps, and the theoretical explanation of why stochastic interpolants help (by breaking bijections) provides much-needed understanding of an empirically observed phenomenon.
- The experimental design is thoughtful, particularly the synthetic Gaussian experiments that allow for analytical verification of theoretical predictions. The CIFAR-10 forward/backward asymmetry analysis provides interesting practical insights about sampling stability.

# Weaknesses
- My main concern is the limited experimental scope. While the Gaussian experiments are convincing, only CIFAR-10 is used for real-world validation. Given the paper's claims about fundamental limitations of flow models, I would expect evaluation on additional datasets to strengthen the generalizability argument. However, given the theoretical nature of this paper, I think this is more of a minor concern.

- The practical impact feels somewhat incremental. While the theoretical understanding is valuable, stochastic interpolants were already used in practice (SBM), and the main practical recommendation (use noise) was already known. The gradient variance diagnostic, while theoretically interesting, doesn't clearly translate to improved training procedures.

- From a clarity perspective, some technical sections are quite dense. The connection between gradient variance and actual transport quality could be made more intuitive. Figure 2's schematic helps but could be more detailed about the mechanisms involved.

---

> ### Author Rebuttal · Authors · 2025-07-29
>
> We thank Reviewer yZa2 for the thoughtful and constructive feedback. We're glad you found the theoretical contributions and gradient variance analysis compelling. Your comments helped us clarify the connection between our results and recent developments, and motivated us to conduct additional experiments on CelebA to broaden the empirical support.
>
> ---
> ## Addressing Weaknesses
>
> > "is the limited experimental scope"
>
> We now include new experiments on CelebA OT pairs from [1] that directly validate Proposition 2 in both adversarial and ReFlow-style settings (see the end of the rebuttal for details).
>
> > "Practical impact feels somewhat incremental."
>
> While adding noise is not novel, our contribution lies in bridging idealized Rectified Flows (infinite data) with practical settings (finite data), where failure modes emerge:
> - no convergence to straight paths after two iterations (Counterexample 1),
> - skipping intersections at inference (Fig. 6),
> - vector fields minimizing both loss and variance reduce to memorization (Prop. 2).
> The noise addition is not our contribution per se, but rather a principled choice aligned with the theory: Prop. 2 holds only under deterministic (noiseless) interpolants.
>
>
> >  The connection between gradient variance and actual transport quality could be made more intuitive.
>
> We will strengthen the paper’s narrative thread by clarifying the following schema, both textually and through an improved Figure 1:
>
> (A) Lemma 1 (Gaussian closed-form) → Prop. 1 (grad. variance bounds)
> (B) Gaussian experiments
> (A)+(B) ⇒ Conclusions (low grad. ≠ OT; reflow skips intersection) → (generalizes) → Prop. 2 (finite data).
>
> ## Questions
>
> > “Provide additional results…”
>
> We provide additional results on CelebA (higher resolution images) OT pairs from [2]
>
> > Gradient Variance as a Tool (increase Practicality)
>
> We agree that utilizing gradient variance as a diagnostic tool could offer a more constructive perspective on the paper. One direction (Fig. 3) is using gradient variance to tune the noise scale during training. For example, one could identify the point at which the gradient variance of a model trained with noise (e.g., SBM) becomes lower—around minima—than that of a model trained with noiseless interpolants. We view this as a separate line of inquiry.
>
> Our motivation for studying gradient variance over time stems from how the term “intersections” has been heavily emphasized in the Rectified Flow literature. In practice, however, intersections almost never occur in high-dimensional spaces with random samples (as formally proven in Proposition 3, Appendix, p. 25). Within this context, gradient variance provides a more meaningful lens for analysis than the geometry of interpolant intersections.
>
> > “Comparison, consistency models…”
>
> Although we did not directly evaluate consistency models in this work, prior research suggests they avoid many of the failure modes we observe in Rectified Flows. Consistency models train a one/k-step mapping, avoiding iterative dynamics that can lead to memorization, but even worse, to degradation of the target as more iterations are put in place.
>
> ## New Experiments on CelebA
>
> To empirically validate Proposition 2, we evaluate Conditional Flow Matching (CFM) on CelebA using ground-truth OT pairings from Korotin et al. [1]. Our goal is to test the central claim: when trained with deterministic interpolants, CFM is optimized to memorize its input pairings.
>
> To test this, we compare:
>
> - **CFM (σ = 0):** trained with deterministic interpolants
>   $x_t = (1 - t) x_0 + t x_1$
>
> - **CFM (σ = 0.05):** trained with slightly noisy interpolants
>   $x_t = (1 - t) x_0 + t x_1 + \sqrt{t(1 - t)} \, \sigma \epsilon$
>
> The small amount of noise ($\sigma = 0.05$) is not introduced for performance reasons but to break the injectivity between $(x_t, t)$ and $(x_0, x_1)$—disabling the memorization pathway outlined in Proposition 2. Both models use identical U-Net architectures.
>
> ---
>
> ### Adversarial Pairings: CFM Optimizes for Memorization with Deterministic Interpolants
>
> **Setup:** To directly test whether CFM with deterministic interpolants ($\sigma = 0$) is optimized to memorize suboptimal pairings, we construct an adversarial dataset:
>
> - We start from ground-truth OT pairs $(x_0, x_1)$ derived from the benchmark of Korotin et al. [2].
> - We shuffle the targets $(x_1)$, breaking the optimal structure and producing random pairings $(x_0, S(x_1))$ with no coherent transport semantics.
>
> We compare CFM (σ = 0) and CFM (σ = 0.05) on $(x_0, S(x_1))$.
>
> | Dataset     | Analysis       | Metric         | CFM (σ = 0.05) | CFM (σ = 0.0) | Advantage                         |
> | ----------- | -------------- | -------------- | -------------- | ------------- | --------------------------------- |
> | 5K Samples  | Generalization | L2 to True OT  | 34.25 ± 7.54   | 50.40 ± 16.73 | CFM (σ = 0.05) generalizes better |
> | 5K Samples  | Memorization   | L2 to Shuffled | 55.02 ± 16.51  | 28.57 ± 5.49  | CFM (σ = 0.0) memorizes more      |
> | 50K Samples | Generalization | L2 to True OT  | 30.05 ± 6.77   | 46.78 ± 14.87 | CFM (σ = 0.05) generalizes better |
> | 50K Samples | Memorization   | L2 to Shuffled | 56.48 ± 18.55  | 45.98 ± 11.85 | CFM (σ = 0.0) memorizes more      |
>
>
> **Evaluation Protocol (Lower is better):**
> We report L2 reconstruction error to  the true OT targets to assess generalization, and  the shuffled training targets to assess memorization.
>
> Since the pairings are deliberately incorrect, a low L2 error to shuffled targets indicates overfitting to suboptimal data. High L2 to these targets signals resistance to memorization.
>
> **Interpretation:**
> The results directly support Proposition 2: CFM with deterministic interpolants ($\sigma = 0$) minimizes loss by tightly memorizing its training targets, even when they are arbitrarily shuffled and non-optimal. In contrast, CFM with small noise ($\sigma = 0.05$) resists this memorization trap and generalizes significantly better to the true OT targets.
>
> **Extra comments:**
> - Memorization decreases with larger datasets, as expected, since model capacity is fixed, it becomes harder to memorize all pairings in the 50K setting.
> - However, this does not contradict Proposition 2: the optimization still favors memorization when possible. Larger datasets simply make perfect memorization less feasible, not less likely as an objective.
>
> ---
>
> ### Simulated 1-ReFlow: CFM Memorizes Its Own Outputs
>
> **Setup:** To evaluate whether iterative flow models suffer from self-reinforcing memorization, we simulate a 1-step ReFlow scenario—a core case of Proposition 2.
>
> - **Initial training:** We train a base CFM model ($\sigma = 0$) on random, non-OT pairings.
> - **Integration:** Using this model, we generate new endpoints $(x_0, x^1)$ via ODE integration.
>
> We compare CFM (σ = 0) and CFM (σ = 0.05) on $(x_0, x^1)$.
>
>
> | Dataset     | Analysis       | Metric        | CFM (σ = 0.05) | CFM (σ = 0.0) | Advantage                         |
> | ----------- | -------------- | ------------- | -------------- | ------------- | --------------------------------- |
> | 5K Samples  | Generalization | L2 to True OT | 31.35 ± 7.38   | 43.35 ± 14.21 | CFM (σ = 0.05) generalizes better |
> | 5K Samples  | Memorization   | L2 to $x^1$     | 25.08 ± 8.59   | 8.63 ± 1.76   | CFM (σ = 0.0) memorizes more      |
> | 50K Samples | Generalization | L2 to True OT | 32.54 ± 8.86   | 38.16 ± 11.37 | CFM (σ = 0.05) generalizes better |
> | 50K Samples | Memorization   | L2 to $x^1$     | 16.29 ± 4.82   | 12.76 ± 4.01  | CFM (σ = 0.0) memorizes more      |
>
>
> **Evaluation Protocol (Lower is better):**
> We report L2 reconstruction error to the true OT targets to assess generalization, and the generated training targets $(x_0, x^1)$ to assess memorization. Since the pairings come from a previous CFM model trained on random (non-OT) data, a low L2 to $x^1$ indicates that the model is overfitting to flawed past outputs. High error rates to these targets signal that the model is resisting memorization and potentially correcting earlier mistakes.
>
> **Interpretation:**
> The results in the table provide further support for Proposition 2: CFM with deterministic interpolants ($\sigma = 0$) is highly effective at memorizing the training couplings, even when those pairings are suboptimal and self-generated. In contrast, CFM ($\sigma = 0.05$), despite only a small amount of noise, avoids this memorization trap and learns to generalize toward the true OT map, despite never being directly trained on it.
>
> ---
>
> ### References
>
> [1] Korotin, Alexander, et al. *Do neural optimal transport solvers work? A continuous Wasserstein-2 benchmark.* NeurIPS 34 (2021): 14593–14605.

---

### Official Review · Reviewer_vzJd · 2025-07-03

**Clarity:** 3
**Significance:** 3
**Originality:** 3
**Rating:** 5
**Confidence:** 3

**Summary:**

This paper proposes multiple theoretical results related to vector field learning.  They show that various ML architectures cannot approximate an optimal transport vector field without error. Next, they assert that low loss gradient variance does not necessarily correspond to an optimal learned vector field.  They provide a series of observations about the impact that stochastic interpolants have on the gradient variance, as compared to deterministic interpolants.  Finally, they present theoretical results on rectified flow models, showing that deterministic interpolants are susceptible to memorization.

**Questions:**

- In L137, it is not clear to me why rotation of the vector field is introduced.  There is no explanation or background provided in the text.  Can you please elaborate on why this is included?
- What does Lemma 1 have to do with the central question that the paper is posing (i.e. what does this tell us about the optimality of vector fields)?  The lemma states that various ML architectures cannot perfectly approximate an optimal vector field — could you please clarify why this is relevant?
- Proposition 1 — it’s unclear to me what the significance of this proposition is.  Bounds are derived for the variance of gradient.  However, the takeaways offered in the text (L162-L166) almost seem to assert that the variance of the gradient does not reliably indicate transport optimality.  Then what is the insight?
- Section 4 — what does this section have to do with analysis of the gradient variance?

**Ethical Concerns:**

["NO or VERY MINOR ethics concerns only"]

**Final Justification:**

In their rebuttals, the authors have provided both additional empirical studies, as well as a detailed elaboration of what they plan to revise/add in order to strengthen the overall narrative. I believe this will make the paper much more readable, and hence more effectively reach its intended audience. My concerns are addressed, therefore I have decided to raise my initial rating by two points.

**Limitations:**

- Results would be more convincing from an empirical perspective if larger, more complicated datasets were used beyond CIFAR-10.

**Paper Formatting Concerns:**

- The writing style of this paper tends to assume that the reader should “just get” why their problem is important, which severely limits its relevance to a broader audience.  Incorporating more connection between the multiple theoretical results, and even a limited discussion of implications on applications, would improve the quality of the paper and make it more readable.

**Quality:**

3

**Strengths And Weaknesses:**

__Strengths:__
- The paper is written with mathematical rigor in definitions and derivations.
- The idea of employing variance of the loss gradient as a “diagnostic” for measuring the faithfulness of learned vector fields to the true optimal transport is creative.
- Insightful result that the introduction of noise via stochastic interpolants can ameliorate certain issues such as stagnation.

__Weaknesses:__
- The significance of the paper’s main results are not clear.  While the theoretical exposition is thorough, the paper would benefit from a more assertive stance on what these multiple results collectively signify.  After reading this paper, I do not come away feeling a greater sense of insight about the question posed in the title of this work.
- Empirical results are evaluated only on a limited dataset (CIFAR-10).  Additional empirical validation on larger, more diverse data domains would highlight the findings more.
- There is an overall lack of “connective tissue” in the writing.  Lemmas and propositions are offered without enough context, and there is very little flow between sections 3 and 4 (and their subsections).  It feels like an ad hoc assortment of observations rather than an intentionally constructed set of arguments supporting a central thesis.

---

> ### Author Rebuttal · Authors · 2025-07-28
>
> We thank the reviewer for the thoughtful feedback and for highlighting the need for clearer conceptual connections. In our revision, we will improve the paper’s structure and include new CelebA experiments (which are also detailed later in this rebuttal) that further validate our theoretical findings.
>
> ---
> ## Addressing Weaknesses
> >  "significance of the paper’s main results are not clear." ,   "The writing style [..] tends to assume that the reader should 'just get' why their problem is important"
>
> Our paper sheds light on the underlying causes of common failure modes behind the state-of-the-art fixed-point generative models. Our main claim is that in noiseless transport settings, deterministic pairings and interpolants, low gradient variance does not imply $W_2$ optimality. This challenges three common assumptions:
>
> 1. Gradient variance arises from interpolant intersections [3, 4]
> 2. ReFlow [5] corrects such intersections via iteration
> 3. Convergence of ReFlow is well-understood [6]
>
> While deterministic ReFlow can straighten paths, it lacks convergence guarantees. A now-retracted claim (with the authors highlighting flaws in the proof of this claim) [1] suggested two iterations suffice; we show that even after two steps, ReFlow may fail to yield injective couplings  (Counterexample 1).
>
> Building on this, we reframed Rectified Flows as an iterative process on finite datasets, as encountered in practice. This revealed that due to discretization and limited data, vector fields can skip over interpolant intersections (Figure 6), failing to correct suboptimal paths. Rather than converging to the optimal transport solution, repeated rectification can lead to memorization: the model simply reproduces the observed pairings, regardless of their quality. We formalize this in Proposition 2, showing that ReFlow can achieve zero loss and gradient variance by memorizing suboptimal deterministic couplings.
>
> >  "Additional empirical validation"
>
> We now include new experiments on CelebA OT pairs from [2] that directly validate Proposition 2 in both adversarial and ReFlow-style settings.
>
> > an overall lack of “connective tissue” in the writing.
>
> We acknowledge the reviewer’s concern, and will strengthen the paper’s narrative thread by clarifying the following schema, both textually and through an improved Figure 1:
>
> (A) Lemma 1 (Gaussian closed-form) → Prop. 1 (grad. variance bounds)
> (B) Gaussian experiments
> (A)+(B) ⇒ Conclusions (low grad. ≠ OT; reflow skips intersection) → (generalizes) → Prop. 2 (finite data).
>
> ## Questions
>
> > Rotations in L137:
>
> The role of rotation (L137) is to introduce a formulation for rotational vector field, which we then study the variance of in Proposition 1.
>
> > What Lemma 1 has to do with the central question that the paper is posing:
>
> The main purpose of  Lemma 1  is to give the closed-form OT vector field formula on which Proposition 1 builds. The extra fact about exact representation is not necessary in the proof of the Proposition and could be introduced as a following remark.
>
> >  Significance of this proposition:
>
>  Proposition 1  supports our empirical findings, which show that low gradient variance does not imply optimality.
>
> > Section 4 relation to the analysis:
>
> Section 4 extends our findings beyond Gaussians. Proposition 2 proves that for deterministic pairings, CFM can reach zero loss and gradient variance by memorizing, even when the vector field is not optimal.
>
> This supports our central claim:  low gradient variance is not a reliable proxy for transport quality. We clarify this thread throughout the paper.
>
> ---
>
> ## New Experiments on CelebA
>
> To empirically validate Proposition 2, we evaluate Conditional Flow Matching (CFM) on CelebA using ground-truth OT pairings from Korotin et al. [2]. Our goal is to test the central claim: when trained with deterministic interpolants, CFM is optimized to memorize its input pairings.
>
> To test this, we compare:
>
> - **CFM (σ = 0):** trained with deterministic interpolants
>   $x_t = (1 - t) x_0 + t x_1$
>
> - **CFM (σ = 0.05):** trained with slightly noisy interpolants
>   $x_t = (1 - t) x_0 + t x_1 + \sqrt{t(1 - t)} \, \sigma \epsilon$
>
> The small amount of noise ($\sigma = 0.05$) is not introduced for performance reasons but to break the injectivity between $(x_t, t)$ and $(x_0, x_1)$—disabling the memorization pathway outlined in Proposition 2. Both models use identical U-Net architectures.
>
> ---
>
> ### Adversarial Pairings: CFM Optimizes for Memorization with Deterministic Interpolants
>
> **Setup:** To directly test whether CFM with deterministic interpolants ($\sigma = 0$) is optimized to memorize suboptimal pairings, we construct an adversarial dataset:
>
> - We start from ground-truth OT pairs $(x_0, x_1)$ derived from the benchmark of Korotin et al. [2].
> - We shuffle the targets $(x_1)$, breaking the optimal structure and producing random pairings $(x_0, S(x_1))$ with no coherent transport semantics.
>
> We compare CFM (σ = 0) and CFM (σ = 0.05) on $(x_0, S(x_1))$.
>
> | Dataset     | Analysis       | Metric         | CFM (σ = 0.05) | CFM (σ = 0.0) | Advantage                         |
> | ----------- | -------------- | -------------- | -------------- | ------------- | --------------------------------- |
> | 5K Samples  | Generalization | L2 to True OT  | 34.25 ± 7.54   | 50.40 ± 16.73 | CFM (σ = 0.05) generalizes better |
> | 5K Samples  | Memorization   | L2 to Shuffled | 55.02 ± 16.51  | 28.57 ± 5.49  | CFM (σ = 0.0) memorizes more      |
> | 50K Samples | Generalization | L2 to True OT  | 30.05 ± 6.77   | 46.78 ± 14.87 | CFM (σ = 0.05) generalizes better |
> | 50K Samples | Memorization   | L2 to Shuffled | 56.48 ± 18.55  | 45.98 ± 11.85 | CFM (σ = 0.0) memorizes more      |
>
>
> **Evaluation Protocol (Lower is better):**
> We report L2 reconstruction error to  the true OT targets to assess generalization, and  the shuffled training targets to assess memorization.
>
> Since the pairings are deliberately incorrect, a low L2 error to shuffled targets indicates overfitting to suboptimal data. High L2 to these targets signals resistance to memorization.
>
> **Interpretation:**
> The results directly support Proposition 2: CFM with deterministic interpolants ($\sigma = 0$) minimizes loss by tightly memorizing its training targets, even when they are arbitrarily shuffled and non-optimal. In contrast, CFM with small noise ($\sigma = 0.05$) resists this memorization trap and generalizes significantly better to the true OT targets.
>
> **Extra comments:**
> - Memorization decreases with larger datasets, as expected, since model capacity is fixed, it becomes harder to memorize all pairings in the 50K setting.
> - However, this does not contradict Proposition 2: the optimization still favors memorization when possible. Larger datasets simply make perfect memorization less feasible, not less likely as an objective.
>
> ---
>
> ### Simulated 1-ReFlow: CFM Memorizes Its Own Outputs
>
> **Setup:** To evaluate whether iterative flow models suffer from self-reinforcing memorization, we simulate a 1-step ReFlow scenario—a core case of Proposition 2.
>
> - **Initial training:** We train a base CFM model ($\sigma = 0$) on random, non-OT pairings.
> - **Integration:** Using this model, we generate new endpoints $(x_0, x^1)$ via ODE integration.
>
> We compare CFM (σ = 0) and CFM (σ = 0.05) on $(x_0, x^1)$.
>
>
> | Dataset     | Analysis       | Metric        | CFM (σ = 0.05) | CFM (σ = 0.0) | Advantage                         |
> | ----------- | -------------- | ------------- | -------------- | ------------- | --------------------------------- |
> | 5K Samples  | Generalization | L2 to True OT | 31.35 ± 7.38   | 43.35 ± 14.21 | CFM (σ = 0.05) generalizes better |
> | 5K Samples  | Memorization   | L2 to $x^1$     | 25.08 ± 8.59   | 8.63 ± 1.76   | CFM (σ = 0.0) memorizes more      |
> | 50K Samples | Generalization | L2 to True OT | 32.54 ± 8.86   | 38.16 ± 11.37 | CFM (σ = 0.05) generalizes better |
> | 50K Samples | Memorization   | L2 to $x^1$     | 16.29 ± 4.82   | 12.76 ± 4.01  | CFM (σ = 0.0) memorizes more      |
>
>
> **Evaluation Protocol (Lower is better):**
> We report L2 reconstruction error to the true OT targets to assess generalization, and the generated training targets $(x_0, x^1)$ to assess memorization. Since the pairings come from a previous CFM model trained on random (non-OT) data, a low L2 to $x^1$ indicates that the model is overfitting to flawed past outputs. High error rates to these targets signal that the model is resisting memorization and potentially correcting earlier mistakes.
>
> **Interpretation:**
> The results in the table provide further support for Proposition 2: CFM with deterministic interpolants ($\sigma = 0$) is highly effective at memorizing the training couplings, even when those pairings are suboptimal and self-generated. In contrast, CFM ($\sigma = 0.05$)—despite only a small amount of noise—avoids this memorization trap and learns to generalize toward the true OT map, despite never being directly trained on it.
>
> ---
>
> ### References
>
> [1] Saptarshi Roy, Vansh Bansal, Purnamrita Sarkar, and Alessandro Rinaldo. *2-rectifications are enough for straight flows: A theoretical insight into Wasserstein convergence.*
>
> [2] Korotin, Alexander, et al. *Do neural optimal transport solvers work? A continuous Wasserstein-2 benchmark.* NeurIPS 34 (2021): 14593–14605.
>
> [3] Fjelde, T., Mathieu, E., and Dutordoir, V. *An Introduction to Flow Matching.* MLG Blog, January 2024.
>
> [4] Gagneux, A., Martin, S., Emonet, R., Bertrand, Q., and Massias, M. *A Visual Dive into Conditional Flow Matching.* ICLR Blogposts, April 2025.
>
> [5] Liu, Xingchao, Chengyue Gong, and Qiang Liu. *Flow straight and fast: Learning to generate and transfer data with rectified flow.* arXiv:2209.03003 (2022).
>
> [6] Lee, Sangyun, Zinan Lin, and Giulia Fanti. *Improving the training of rectified flows.* NeurIPS 37 (2024): 63082–63109.

---

> > ### Comment · Reviewer_vzJd · 2025-08-06
> >
> > Thank you to the authors for your rebuttal.  I appreciate the additional experiments provided by the authors (CelebA), and the discussion gives me some further clarity about the significance of the theoretical findings.  However, I still think there are serious issues regarding clarity of the central argument (i.e., that contrary to existing literature, gradient variance is a poor indicator of learned vector field optimality).  While the individual arguments make sense to me, I still feel that the paper is fairly roundabout in its assertions, asking the reader to piece together the authors' arguments.  While the theoretical basis and experimental methodology of this work are solid, the writing style (which includes the authors' rebuttal) remains a significant barrier to communicating the scientific relevance to a broad audience.
> >
> > ________________________________________________________________
> > ## Question:
> >
> > ```
> > We acknowledge the reviewer’s concern, and will strengthen the paper’s narrative thread by clarifying the following schema, both textually and through an improved Figure 1:
> >
> > (A) Lemma 1 (Gaussian closed-form) → Prop. 1 (grad. variance bounds)
> > (B) Gaussian experiments
> > (A)+(B) ⇒ Conclusions (low grad. ≠ OT; reflow skips intersection) → (generalizes) → Prop. 2 (finite data).
> > ```
> > While I believe this will be a valuable improvement, I am not convinced that the provided response sufficiently addresses my concerns.  Could the authors please provide the the actual clarifications you are planning to provide regarding this "schema"? I.e. in understandable prose -- the current abbreviated description is confusingly terse.
> > - For each of these points (A, B, A+B), what exactly will you clarify for the reader, and how will you provide connective tissue (please state this explicitly).
> > - What are you planning to improve about Figure 1?  As is, the ad hoc structure of this diagram completely obscures the takeaways it is intended to showcase.
> > ________________________________________________________________
> >
> > I am open to considering a score change pending further discussion of these remaining concerns, but as is (including the 1st rebuttal), I still lean toward rejection rather than acceptance.

---

> > > ### Author Response · Authors · 2025-08-07
> > > **Readability Improvements**
> > >
> > > > For each of these points (A, B, A+B), what exactly will you clarify for the reader, and how will you provide connective tissue (please state this explicitly).
> > >
> > > We appreciate the reviewer’s continued engagement, and agree that our earlier response was too abbreviated. Below, we provide a clearer and more structured breakdown of how we will revise the narrative in the paper to reflect the schema (A, B, A+B), including explicit connective commentary and concrete revision plans.
> > >
> > > ---
> > >
> > > ### (A) Theoretical Basis: When is Gradient Variance Informative? (Section 3)
> > >
> > > We will explicitly clarify that the relationship between gradient variance and vector field optimality depends on the training regime. We distinguish:
> > >
> > > - Stochastic regimes (e.g., score-based models and CFM): where training is performed over random couplings.
> > > - Deterministic regimes (e.g., ReFlow(k > 1)): where training interpolants are fixed after the first step.
> > >
> > > We will clarify that the relevance of gradient variance depends crucially on which regime one is in.
> > >
> > > Connective tissue: We will make the following logical progression more explicit in the revised text:
> > >
> > > - Lemma 1 derives closed-form vector fields for OT between Gaussians.
> > > - Proposition 1 then uses this to construct explicit counterexamples: vector fields with low gradient variance that are not OT.
> > >
> > > We will add a summary paragraph within Section 3 to walk the reader through this progression step-by-step, emphasizing why the theoretical result undercuts the use of gradient variance as a general diagnostic.
> > >
> > > ---
> > >
> > > ### (B) Empirical Support: Gaussian Experiments (Section 3.1)
> > >
> > > Clarification: We will reinforce the alignment between theory and experiment: the counterexamples predicted in Proposition 1 are concretely observed in our controlled Gaussian setting. Solutions with low variance visibly differ from the OT map.
> > >
> > > Connective tissue: We will revise the start of Section 3.1 to explicitly reference Proposition 1 and explain how the empirical plots are constructed to validate its predictions. We will also clarify that this section is not just illustrative but forms a bridge from theoretical to practical settings.
> > >
> > > ---
> > >
> > > ### (A+B) Implications for ReFlow and Real Data (Section 4)
> > >
> > > Clarification: Section 4 generalizes the analysis to finite datasets, especially focusing on ReFlow. We will emphasize that:
> > >
> > > - ReFlow achieves low gradient variance due to its deterministic training regime, but
> > > - The resulting vector fields can be suboptimal or misaligned with OT,
> > > - And Proposition 2 explains why ReFlow cannot recover from these early misalignments.
> > >
> > > Connective tissue: We will explicitly state at the start of Section 4 that this is the finite-data extension of Section 3’s insight. We will introduce Proposition 2 with a recap of the earlier Gaussian findings to remind the reader why low variance may not be trustworthy, and draw a parallel between deterministic ReFlow and the Gaussian counterexamples.
> > >
> > > ---
> > >
> > > ### Revision Strategy to Improve Narrative Clarity
> > >
> > > To make these contributions easier to follow, we plan the following textual and visual revisions:
> > >
> > > - We will add bridging paragraphs that explicitly reference earlier results (e.g., “recall from Proposition 1 that...”) to guide the reader through the chain of logic.
> > > - We will move Section 3.2 to the Appendix to streamline the mainline narrative in Section 3.
> > > - We will reframe Section 4 as a generalization of Section 3, tying it more clearly to the Rectified Flows framework.
> > > - We will clarify how Lemma 1 informs both the theory (Prop. 1) and experiments (Sec 3.1), so that it doesn’t appear isolated.
> > > - We will emphasize in Section 4.3 that introducing stochasticity breaks ReFlow’s failure mode, connecting back to the opening contrast between training regimes.
> > >
> > > ---
> > >
> > > > What are you planning to improve about Figure 1? As is, the ad hoc structure of this diagram completely obscures the takeaways it is intended to showcase.
> > >
> > > We agree that the current structure is unclear. The revised Figure 1 will be a structured flowchart, explicitly mirroring the paper’s logic:
> > >
> > > 1. Is gradient variance reliable?
> > >    → Yes, in stochastic regimes (SBM, CFM).
> > >    → No, in deterministic regimes (ReFlow(k > 1)).
> > >
> > > 2. Why not?
> > >    → Theoretical result (Prop. 1)
> > >    → Empirical Gaussian validation
> > >    → Real-world example: ReFlow failure (Prop. 2, CelebA)
> > >
> > > 3. What helps?
> > >    → Injecting stochasticity (e.g., CFM+SI)
> > >    → Prevents memorization, recovers better couplings
> > >
> > > Each node will be labeled with section numbers and key results to tightly couple the figure to the paper’s structure and claims.

---

> > > > ### Comment · Reviewer_vzJd · 2025-08-07
> > > >
> > > > Thank you authors for providing this detailed elaboration of what you plan to revise/add in order to strengthen the overall narrative.  I believe this will make the paper much more readable, and hence more effectively reach its intended audience.  My concerns are addressed, and I am looking forward to reading the final version of the paper.  I have decided to raise my rating by two points.

---

### Note · Authors · 2025-08-12

We thank the reviewers and the Area Chair for their engagement throughout the process. Following the discussion, we added CelebA experiments to support our findings. In the final version, we will revise the manuscript to improve clarity, restructure Sections 3 and 4, move Section 3.2 to the appendix, and replace Figure 1 with a clearer schematic. We appreciate the feedback and will reflect it in the revision.

---

### Decision · Program_Chairs · 2025-09-17

**Decision:**

Accept (spotlight)

**Comment:**

The paper analyzes failures of state-of-the-art fixed-point generative models—Rectified Flows and Schrödinger Bridge Matching—by introducing a gradient-variance diagnostic that quantifies transport error after integration. They also show that even low-variance (well-aligned) interpolants can still produce large transport errors. The authors establish architectural limitations, proving that standard MLP/CNN architectures cannot exactly realize Gaussian-to-Gaussian transport. It further shows that with noiseless interpolants, the fixed-point procedure either stagnates on straight pairings or memorizes deterministic non-straight pairings rather than approaching optimal transport, and proves the existence of such vector fields. The reviewers praise the strong theoretical results, clear counterexamples, and novelty of the introduced diagnostic. The authors should address concerns about the limited empirical scope by including results for additional datasets. The authors should also include the results using the FID-based benchmark in the revised manuscript, update it with the clarifications provided in the rebuttal, and incorporate the reviewers' comments to improve clarity.